# Med-Real2Sim: Non-Invasive Medical Digital Twins using Physics-Informed Self-Supervised Learning

**Keying Kuang**[*]
UC Berkeley

**Frances Dean**[*]
UC Berkeley & UCSF

**Jack B. Jedlicki**[*]
University of Barcelona

**David Ouyang**
Cedars Sinai

**Anthony Philippakis**[†]
Google Ventures

**David Sontag**[†]
MIT

**Ahmed Alaa**[†]
UC Berkeley & UCSF

## Abstract

A digital twin is a virtual replica of a real-world physical phenomena that uses mathematical modeling to characterize and simulate its defining features. By constructing digital twins for disease processes, we can perform in-silico simulations that mimic patients' health conditions and counterfactual outcomes under hypothetical interventions in a virtual setting. This eliminates the need for invasive procedures or uncertain treatment decisions. In this paper, we propose a method to identify digital twin model parameters using only noninvasive patient health data. We approach the digital twin modeling as a *composite inverse problem*, and observe that its structure resembles pretraining and finetuning in self-supervised learning (SSL). Leveraging this, we introduce a *physics-informed SSL* algorithm that initially pretrains a neural network on the pretext task of learning a differentiable simulator of a physiological process. Subsequently, the model is trained to reconstruct physiological measurements from noninvasive modalities while being constrained by the physical equations learned in pretraining. We apply our method to identify digital twins of cardiac hemodynamics using noninvasive echocardiogram videos, and demonstrate its utility in unsupervised disease detection and in-silico clinical trials.

## 1 Introduction

With increasing health data availability, there is excitement about refining physics-based models of human body systems to be patient specific, as personalized physical models can provide the basis for in-silico experiments and timely diagnosis (27; 31; 55; 25). This personalized vision of healthcare has given rise to virtual patients constructed with data-tuned models, known as *digital twins*. Originally an engineering concept, digital twins have recently been realized as a resource in healthcare (6; 4). The concept combines data-driven approaches with mechanistic or simulation techniques, serving as a bidirectional map between real data and simulations. Medical digital twins have been applied and developed broadly in applications ranging from cellular mechanics (24) to the development of whole body and human digital twins (50; 76), with aims ranging from tailoring cardiovascular interventions (30) to agent-based trauma care management systems (14). Both academic and industry efforts are prolific with digital twins having been developed for various medical problems (9; 69; 74; 13; 22; 70).

In this paper, we propose a framework for identifying patient-level digital twins using noninvasive medical images. We focus on scenarios where identifying a patient's digital twin from noninvasive data allows us to simulate other physiological parameters that typically require invasive measurement, such as through catheterization. Such an approach assumes that there is a mapping from rich non-invasive imaging data to patient-specific mechanistic models of physiology; such a mapping is not known to scientists humans but can be learned from data. This hypothesis is supported by previous experimental work (2; 64; 43). However, the challenge lies in the absence of datasets containing both

38th Conference on Neural Information Processing Systems (NeurIPS 2024).

noninvasive data and the corresponding parameters of the underlying physics-based physiological model, hence the problem cannot be solved using standard supervised learning. To this end, we propose a new class of physics-informed neural networks (PINNs) (60) to incorporate inductive biases informed by mechanistic models of patient physiology into neural network architectures.

**Summary of contributions.** We study obtaining latent parameters of a physics-based model from image measurements directly as an inverse problem. Learning patient specific parameters for a physics-based model from a complete description of patient physical states and a known forward model is an inverse problem. This task can be modeled using, for example, a PINN approach (71; 39; 29). The process of obtaining patient physical states from non-invasive medical imaging is a second inverse problem with an *unknown* forward model. Obtaining patient specific latent parameters from non-invasive medical imaging directly combines these two inverse problems and could be modeled by learning the unknown process components. Large numbers of labeled data pairs of

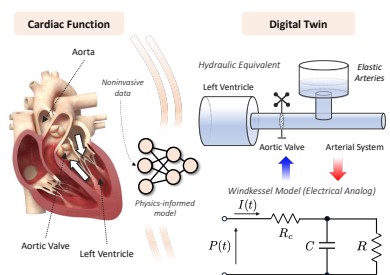

Figure 1: **Digital twins for cardiac hemodynamics**. Left: Illustration of the cardiovascular system. Right: Digital twin models of cardiac hemodynamics based on hydraulic or electric representations.

images and parameters for the full inverse task would need to be obtained from invasive procedures to use supervised learning. Instead, we propose a training approach that structurally resembles self-supervised learning (SSL), and leverages labeled data pairs of images and non-invasively measurable patient physical states. We first train a neural network using supervised learning with *synthetic* data to approximate the forward physics-based model as a pre-text task. We then freeze this network to fine-tune the learning of a solution to the composite inverse problem from the labeled measurable data pairs. Our methods are then extendable to include exogenous intervention parameters for the development of in-silico experiments. We call this framework "Med-Real2Sim" since it learns virtual simulators of patient physiology on the basis of real (noninvasive) data.

We apply our methodology in the setting of predicting patient specific cardiac pressure-volume loops from echocardiography. The relationship between left-ventricle pressure and volume is an important description of cardiac function (36). Pressure-volume catheter measurements are highly invasive, challenging their routine clinical use (8; 1; 77). Cardiac ejection fraction, the ratio between left-ventricular stroke volume and end-diastolic volume, while more routinely measured as a proxy for cardiac function, does not take into account cardiac pressure, preventing diagnosis of many severe cardiac conditions and providing motivation to develop digital twins for pressure-volume relations (80; 45; 59). Cardiovascular pressure and volume are commonly modeled using hydraulic analogy models, representing the hydraulic system by electronic circuits, characterized by lumped parameters corresponding to system attributes. Systems of differential equations describe the relationships between circuit parameters and volume and pressure states. A model developed in (66) relates left-ventricular pressure and volume with and without the addition of a left-ventricular assistance device (LVAD). We train a neural network to learn to map between parameters and solution states of this model and use this learning to fine-tune a convolutional neural network that learns patient specific model parameters using echocardiography frames.

## 2 Method

### 2.1 Problem Setup

We consider a physics-based model $\mathcal{M} : \Theta \to \mathcal{X}$ of a physiological process which maps a set of parameters $\theta \in \Theta$ to a set of states $\mathbf{x} \in \mathcal{X}$. The parameters $\theta$ describe the physiological process in an individual patient; each patient $i$ is associated with a patient-specific parameter set $\theta_i$ and physical states $\mathbf{x}_i$. We are interested in a setup where the complete set of parameters and states can only be measured *invasively*, e.g., through a catheterization procedure. We conceptualize the physics-based model $\mathcal{M}(\theta_i)$ as the "Digital Twin" of patient $i$. That is, if we have oracle access to the parameters $\theta_i$ of patient $i$, then we can simulate their physiological states through the following *forward* process:

$$\mathbf{x} = \mathcal{M}(\theta). \tag{1}$$

The model $\mathcal{M}$ is assumed to be known and would typically be defined by a system of differential equations written $f_\theta(\mathbf{x}) = 0$ with a physically interpretable parameter set $\theta$. One can use numerical

methods to solve $f_\theta(\mathbf{x}) = 0$ for $\mathbf{x}$ and construct the solution $\mathcal{M}(\theta_i)$ for each patient parameter set $\theta_i$ but a closed form solution to define the map $\mathcal{M}$ may not exist even though the model is known, such as in the case where $f_\theta = 0$ has no closed form solution. The level of granularity in simulating the (true) physiological process depends on the fidelity of the physics-based model $\mathcal{M}$.

In addition to modeling the natural state of the physiological process, we also consider physics-based models of clinical interventions (e.g., medical devices). To model an interventional process, we consider an augmented version of the physics-based model $\widetilde{\mathcal{M}} : \Theta \times \mathcal{U} \to \mathcal{X}$, which maps the *endogenous* parameters $\theta \in \Theta$ of the patient and an *exogenous* intervention $u \in \mathcal{U}$ to a physiological state $\mathbf{x} \in \mathcal{X}$. The augmented model describes how the intervention $u$ alters the physiological states of a patient with parameters $\theta$—in the causal inference literature, this class of models is known as structural causal models (SCMs) (57).

To identify the model $\mathcal{M}(\theta)$ traditionally and simulate the states $\mathbf{x}$, one would need to conduct a number of invasive procedures for each patient. Here, we assume that we have access to a measurement $\mathbf{y}_i$ for each patient $i$, which reflects some aspects of the underlying physiological process. We assume that $\mathbf{y}$ is acquired through a *non-invasive* procedure that might be conducted routinely in clinical practice (e.g., an electrocardiogram). The relationship between the physiological state $\mathbf{x}$ and the non-invasive measurement $\mathbf{y}$ is given by

$$\mathbf{y} = \mathcal{K}(\mathbf{x}). \tag{2}$$

Thus, by combining (1) and (2), we arrive at the following relation between the parameters of the physics-based model and the non-invasive measurements:

$$\mathbf{y} = \underbrace{\mathcal{K}}_{\text{Unknown}} \circ \underbrace{\mathcal{M}}_{\text{Known}}(\theta) = \mathcal{F}(\theta). \tag{3}$$

That is, the relation between the non-invasive measurements and the physical model parameters is a composition of two functions; an *unknown* map $\mathcal{K}$ that describes the relation between the physiological states and the observed measurements (e.g., the relation between pixels in an imaging modality and underlying health states), and a *known* mathematical model $\mathcal{M}$ that captures the relation between the physiological states and *unknown* parameters $\theta$. **Our goal** is to develop a learning algorithm to identify the digital twin $\mathcal{M}(\theta_i)$ for patient $i$ given their non-invasive measurement $\mathbf{y}_i$.

**Digital Twin Modeling as an Inverse Problem.** Note that obtaining patient-specific parameters $\theta_i$ from physical states $\mathbf{x}_i$ amounts to solving the following inverse problem:

$$\theta = \mathcal{M}^{-1}(\mathbf{x}). \tag{4}$$

However, we do not directly observe the state $\mathbf{x}$, but rather a measurement $\mathbf{y}$ obtained by passing $\mathbf{x}$ through the unknown forward model $\mathcal{K}$. Thus, obtaining the physical parameters from $\mathbf{y}$ entails solving the composite inverse problem given by:

$$\theta = \mathcal{F}^{-1}(\mathbf{y}) = \mathcal{M}^{-1} \circ \mathcal{K}^{-1}(\mathbf{y}). \tag{5}$$

Solving this problem entails two challenges. First, we do not know the forward model $\mathcal{K}$. Second, we cannot directly learn the inverse map $\mathcal{F}^{-1}$ using supervised learning since this requires access to a labeled dataset $\{(\mathbf{y}_i, \theta_i)\}_i$. Obtaining such a dataset would require conducting a large number of invasive procedures, which we seek to avoid.

An alternative way to learn the inverse map $\mathcal{F}^{-1}$ is to break down the composite inverse problem in (5) into separate inverse problems, and only learn the inverse of the unknown map $\mathcal{K}^{-1}$ using examples of the form $\{(\mathbf{y}_i, \mathbf{x}_i)\}_i$ by fixing the map $\mathcal{M}$. If we can construct a mapping representative of $\mathcal{M}$ from any patient parameter set $\theta \in \Theta$ to state $\mathbf{x} \in \mathcal{X}$, then as we can write $\mathcal{K}^{-1}$ as follows:

$$\mathcal{K}^{-1} : \mathcal{Y} \to_{\mathcal{F}^{-1}} \Theta \to_{\mathcal{M}} \mathcal{X} \tag{6}$$

we can learn $\mathcal{F}^{-1}$ using training samples which only train $\mathcal{K}^{-1}$.

As we will ultimately use supervised methods and gradient descent to learn Eq. (6), we need a method for fixing $\mathcal{M}$ that allows for mapping any arbitrary parameter set $\theta$ to corresponding state $\mathbf{x}$. There are several methods in the literature for solving the forward dynamics of $\mathcal{M}$ that we might think of to construct this mapping. These include physics-informed neural networks (PINNs) (60), which train a network to approximate the solution $\mathbf{x}_i$ to $f_{\theta_i}(\mathbf{x}_i) = 0$ thereby solving $\mathcal{M}(\theta_i)$ for a fixed patient parameter set $\theta_i$. Traditional PINNs learn the function $\mathbf{x}_i$ by incorporating the differential

equation $f_{\theta_i} = 0$ as a soft-constraint in the network's training loss alongside a data loss. In the setting of a ordinary differential equations, Neural ODEs (10) train a network to learn the differential equation operator $f_{\theta_i}$ on the states $\mathbf{x}_i$ to solve for $\mathbf{x}_i = \mathcal{M}(\theta_i)$ using differentiable ODE solvers. These methods can learn $\mathcal{M}(\theta_i)$ for a fixed patient parameter set $\theta_i$ and can be compared to the use of a numerical differential equation solvers.

These existent approaches for solving $\mathcal{M}$, however, only give the states $\mathbf{x}$ for a single patient at a time. To learn $\mathcal{K}^{-1}$ by training a network *through* the map $\mathcal{M}$, we need a method that approximates the *map* as otherwise, we would need to train infinitely many separate PINNs or Neural ODEs to cover the entirety of a plausible parameter range potentially traversed in supervised learning of (6).

**Non-Invasive Data Acquisition.** While we do not have direct access to the patient physiological state $\mathbf{x}_i$ to train a model to learn $\mathcal{K}^{-1}$, the noninvasive observation $\mathbf{y}_i$ typically measures physiological quantities that can also be derived from the true state $\mathbf{x}_i$. We assume that $\bar{\mathbf{x}}$ is a physiological variable that can be derived from both $\mathbf{x}$ and $\mathbf{y}$, i.e.,

$$\bar{\mathbf{x}} = m(\mathbf{x}) = g(\mathbf{y}), \tag{7}$$

where $m(.)$ is a known function and $g(.)$ is a labeling procedure, typically conducted manually by a physician or automatically through a built-in algorithm in the data acquisition device. This physiological variable represents a quantity of interest assessed by the noninvasive modality (e.g., breast density in mammography, blood pumping efficiency in ultrasound). These variables establish a connection between the true state $\mathbf{x}$ and the observation $\mathbf{y}$, as they can be both simulated from the physical model and measured based on $\mathbf{y}$.

Given the setup above, our digital twin modeling problem can be formulated as follows. Given a dataset of noninvasive measurements for $n$ patients, $\mathcal{D} = \{\mathbf{y}_i\}_{i=1}^n$, and a physiological variable of interest $\bar{\mathbf{x}}_i = m(\mathbf{x}_i) = g(\mathbf{y}_i)$, our goal is to train a model that can identify the underlying physics-based twin $\mathcal{M}(\theta_{n+1})$ for a new patient $n+1$ based solely on their corresponding noninvasive measurement $\mathbf{y}_{n+1}$. We seek to do this by fixing $\mathcal{M}$ and learning $\mathcal{K}^{-1}$ from the data $\mathcal{D} = \{(\mathbf{y}_i, \bar{\mathbf{x}}_i)\}_{i=1}^n$.

## 2.2 Physics-Informed Self-Supervised Learning

Given the training dataset $\mathcal{D} = \{(\mathbf{y}_i, \bar{\mathbf{x}}_i)\}_{i=1}^n$, a typical supervised learning task is to train a model to predict the physiological variable $\bar{\mathbf{x}}_i$ on the basis of the noninvasive observation $\mathbf{y}_i$ via standard empirical risk minimization (ERM):

$$\widehat{f} = \arg\min_f \tfrac{1}{n} \sum_{i=1}^n \ell(f(\mathbf{y}_i), \bar{\mathbf{x}}_i). \tag{8}$$

The motivation for training the model $\widehat{f}$ is usually to automate the collection of physiological variables from medical images in a clinical workflow (68). These models follow a fully data-driven approach and do not incorporate our knowledge of how the physiological processes being measured function on the biological or physical levels. One could think of the supervised model $\widehat{f}$ as a data-driven model of the function $g(.)$ in (7).

Recall that our goal is to recover the latent physical parameters $\theta$ that underlie the physiological processes that generated the observation $\mathbf{y}$. If we had oracle access to a dataset of the form $\mathcal{D}^* = \{(\mathbf{y}_i, \theta_i)\}_{i=1}^n$, then a supervised solution to identifying the digital twin would be:

$$\widehat{f} = \arg\min_f \tfrac{1}{n} \sum_{i=1}^n \ell(f(\mathbf{y}_i), \theta_i). \tag{9}$$

The solution to (9) provides an approximate solution to the inverse problem in (5). As we lack access to $\mathcal{D}^*$, we can only learn the solution to (5) using the observed dataset $\mathcal{D}$. To this end, we propose a two-step approach for learning the parameters $\theta$ from $\mathbf{y}$, leveraging the structural similarity between the composite inverse problem in (5) and the pretraining/finetuning paradigm in self-supervised learning (SSL). An illustration of the two steps is provided in Figure 2(a).

**Step 1: Physics-Informed Pretext Task.** In this step, we pretrain a neural network to imitate the forward dynamics of the physics-based model, i.e., $\mathbf{x} = \mathcal{M}(\theta)$. We do so by first sampling $\tilde{n}$ synthetic training examples from the physics-based forward dynamics as follows:

$$\tilde{\theta}_j \sim \text{Uniform}(\Theta), \; \tilde{\mathbf{x}}_j = \mathcal{M}(\tilde{\theta}_j), \; 1 \leq j \leq \tilde{n}. \tag{10}$$

We denote this synthetic dataset by $\widetilde{\mathcal{D}} = \{(\tilde{\theta}_j, \tilde{\mathbf{x}}_j)\}_j$. Next, we use this dataset to train a feed forward neural network on the *pretext* task of predicting the patient physiological states from the physical parameters using ERM as follows:

$$\widehat{\phi}_{\mathcal{M}} = \arg\min_{\phi} \tfrac{1}{\tilde{n}} \sum_{j=1}^{\tilde{n}} \ell(\phi(\tilde{\theta}_j), \tilde{\mathbf{x}}_j). \tag{11}$$

This training process distills the true physics-based model, $\mathcal{M}(\theta)$, into the weights of a neural network $\widehat{\phi}_{\mathcal{M}}$. That is, the forward pass of the network will emulate the forward dynamics of the physiological process, i.e., $\mathbf{x} \approx \widehat{\phi}_{\mathcal{M}}(\theta)$.

**Step 2: Physics-Guided Finetuning.** Given the observed dataset $\mathcal{D} = \{(\mathbf{y}_i, \bar{\mathbf{x}}_i)\}_i$, we train another model to predict the physical parameters $\theta$ from the observed measurements $\mathbf{y}$ using the loss function

$$\widehat{\phi}_{\mathcal{F}} = \arg\min_{\phi} \tfrac{1}{n} \sum_{i=1}^{n} \ell(m(\widehat{\phi}_{\mathcal{M}} \circ \phi(\mathbf{y}_i)), \bar{\mathbf{x}}_i). \tag{12}$$

Here, the model pretrained on synthetic data from the physical simulator, $\widehat{\phi}_{\mathcal{M}}$, is frozen and only the model $\widehat{\phi}_{\mathcal{F}}$ is finetuned using real data on $\mathbf{y}$ and $\bar{\mathbf{x}}$. The neural network trained in Step 2, $\widehat{\phi}_{\mathcal{F}}$, represents an approximate solution to the composite inverse problem in (3), i.e., $\widehat{\phi}_{\mathcal{F}}(.) \approx \mathcal{F}^{-1}(.)$. For a new patient $n+1$, we discard the pretrained model $\widehat{\phi}_{\mathcal{M}}$, and use the model $\widehat{\phi}_{\mathcal{F}}$ to predict the patient's digital twin based on their noninvasive measurement $\mathbf{y}_{n+1}$ as follows:

$$\widehat{\theta}_{n+1} = \widehat{\phi}_{\mathcal{F}}(\mathbf{y}_{n+1}), \ \widehat{\mathbf{x}}_{n+1} = \mathcal{M}(\widehat{\theta}_{n+1}). \tag{13}$$

**Interpretation of Physics-Informed SSL.** Our proposed algorithm structurally resembles the SSL paradigm, where, akin to SSL, we decompose our model into a "backbone" architecture and a task-specific "head" (47). However, there are fundamental conceptual differences between standard SSL approaches and ours. In standard SSL, the backbone is a high-capacity model pretrained on a pretext task using unlabeled data to derive a general-purpose representation transferable to many downstream tasks, while the low-capacity head is finetuned for the specific task. In our physics-informed approach, SSL serves to constrain rather than enhance flexibility. We first pretrain the low-capacity head to distill the laws of physics, and then finetune the high-capacity backbone model to learn a mapping that aligns with the frozen head to produce the observed physiological variables $\mathbf{x}$ while respecting the physical laws. In doing so, we force the backbone model to learn a mapping from the measurement $\mathbf{y}$ to the physical parameters $\theta$.

The concept of distilling a physics-based dynamic model into a data-driven one draws from engineering methodologies that develop surrogate models to simplify computational complexity using simulated data (e.g., (58)). This data-driven approach mirrors other supervised learning generative strategies for solving inverse problems or system identification with known dynamics in control theory (51; 78). Importantly, physics-informed SSL scales efficiently with increasingly complex and higher fidelity physical models, as its computational requirements involve one offline process (Eq. (10) and (11)) once the synthetic data is acquired.

# 3 Digital Twins for Cardiovascular Hemodynamics

We apply our physics-informed SSL method to perform patient-specific Pressure–Volume (PV) loop analysis in cardiology using echocardiography (cardiac ultrasound). Left ventricular (LV) PV loops illustrate the LV pressure against LV volume at multiple time points during a single cardiac cycle, providing a comprehensive view of a patient's cardiac function and encoding various hemodynamic parameters like stroke volume, cardiac output, ejection fraction, myocardial contractility, etc., which can be used to diagnose cardiovascular diseases (7). Traditionally, a full characterization of a PV loop requires an invasive PV loop catheterization procedure (35). Our objective is to utilize a noninvasive modality (i.e., echocardiography) to identify a digital twin for each patient, allowing us to simulate their individualized PV loops.

**Physics-based model.** We utilized a lumped-parameter circuit model of cardiac hemodynamics (i.e., blood flow), also known as the Windkessel model (79), as our underlying physical model $\mathcal{M}(\theta)$. Lumped parameter models are based on the analogy between blood flow in arterial systems and the flow of electric current in a circuit (Figure 1). The model comprises interconnected compartments that are equivalent to elements of an electric circuit. Resistances represent the resistance of blood flow

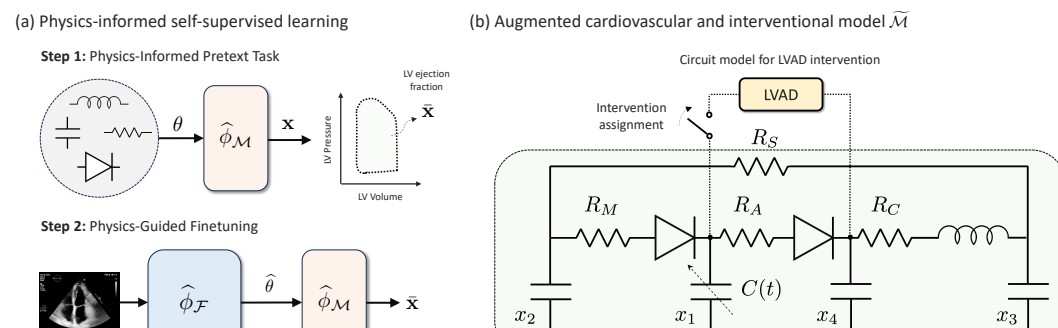

Figure 2: **Illustration of Med-Real2Sim digital twins for cardiovascular hemodynamics.** (a) Pictorial depiction of the two-step physics-informed SSL algorithm proposed in Section 2.2. (b) Five state lumped-parameter electric circuit model of cardiac hemodynamics from (66). Here $\mathbf{x}(t) = [x_1(t), x_2(t), x_3(t), x_4(t), x_5(t)] = [P_{LV}(t), P_{LA}(t), P_A(t), P_{Ao}(t), Q(t)]$ describes the voltages $x_1, x_2, x_3, x_4$ or pressures in the left-ventricle, left atrium, arteries, and aorta, respectively, and total flow $x_5$. The LVAD is modeled through an electric circuit connected to the digital twin via a switch. An LVAD intervention is applied if the switch is closed.

within the blood vessels, accounting for pressure energy losses in the system; capacitors represent the amount of stored stressed blood; diodes describe mitral and aortic valves, and inductances describe the inertial effects due to blood acceleration (83). The blood flow within the system follows the standard Kirchoff's voltage and current laws. A patient's digital twin is represented by an electric circuit with a specific parameter instance $\theta$, where $\theta$ corresponds to the components of the electric circuit. The patient's physiological state $\mathbf{x}$ corresponds to circuit currents and voltages (i.e., blood flow and pressures), which enables simulation of the patient's PV loop in any local heart structure. In this study, we adopt the five-state electric circuit model from (66), illustrated in Figure 2(b).

The cardiac LV pressure $P_{LV}(t)$ and LV volume $V_{LV}(t)$ are related by the elastance function, $E(t) = P_{LV}(t)/(V_{LV}(t) - V_d)$, which is modeled using a closed form in Eq. (15) in the Appendix. Evolution of $\mathbf{x}$ over time $t$ is governed by a system of ODEs derived from current conservation laws (Appendix Eq. (20)). The model is parameterized by a vector $\theta$ corresponding to bounds on the elastance, circuit resistance, capacitance, inductance and initial states (see Appendix A.2). Given the patient-specific parameters $\theta_i$ for the ODE system and elastance function governing cardiac dynamics, we have a unique solution for $\mathbf{x}_i$, i.e., $\mathcal{M}$ is injective (Appendix B).

**Modeling interventions.** A left-ventricular assistance device (LVAD) is a blood pump that helps improve cardiac function in severely ill patients. The effect of implanting an LVAD is modeled in with the addition of one state variable describing blood flow through the device and a tuneable parameter. The impact of LVAD on blood flow can be modeled by attaching an exogenous electric circuit to $\mathcal{M}(\theta)$ to form an augmented model $\widetilde{\mathcal{M}}(\theta)$ (Figure 2(b)). Details on the mathematical model of LVAD are in Appendix A.

**Noninvasive data.** For each patient $i$, we have access to an echocardiogram in the form of ultrasound video data $\mathbf{y}_i$ and a physician labeled measurement of the LV ejection fraction $\bar{\mathbf{x}}_i$. Echocardiography is a widely used noninvasive modality for diagnosing cardiovascular diseases; it can directly provide volumes but not pressure measurements. Our goal is to use the echocardiography clip for a patient to predict their entire PV loop through a fully noninvasive process.

## 4 Experiments

### 4.1 Echocardiography Data

We test our physics-informed SSL (Med-Real2Sim) approach using two echocardiography datasets: EchoNet and CAMUS. The CAMUS dataset (44) consists of 500 fully annotated cardiac ultrasound videos in 2-chamber view, each with LV volume labels for end systole and diastole ($V_{LV}(t_{ES})$ and $V_{LV}(t_{ED})$). The videos were processed by spatial and temporal padding, with standardized 30-frame videos with a resolution of 256×256 pixels. The EchoNet dataset (53) comprises 10,030 apical-4-chamber echocardiography videos from routine clinical care at Stanford University Hospital

also labeled with $V_{LV}(t_{ES})$ and $V_{LV}(t_{ED})$. These videos were cropped, masked, and down-sampled to a resolution of $112\times112$ pixels using cubic interpolation. The CAMUS dataset was split into 450 training samples and 50 validation and testing samples. The EchoNet dataset was partitioned into 7,466 training, 1,288 validation, and 1,276 testing samples.

## 4.2 Simulating individualized PV loops via digital twins

We train a 3D-CNN implemented using Pytorch to output a subset of the model parameters $\theta_i$. The network consists of four convolutional layers, each followed by max pooling and concluded by a convolutional layer with global average pooling and two fully connected layers. We choose seven of the model parameters listed in Table 3 to be learned in our model: mitral valve resistance $R_M$, aortic valve resistance $R_A$, maximum elastance $E_{\max}$, minimum elastance $E_{\min}$, theoretical LV volume at zero pressure $V_d$, start LV volume $V_{LV}(0)$, and heart cycle duration $T_C$. The remainder of the parameters are fixed to literature values (see Table 3). We enforce restrictions on the parameters by normalizing values in our activation layer, which scales parameters using a sigmoid function to ensure parameters fall into realistic bounded ranges (as in Table 3). Parameters $\theta_i$ are then passed to a separately trained fixed-weight feed forward neural network $\widehat{\phi}_{\mathcal{M}}$ to output the labeled values of $\bar{\mathbf{x}}_i$ from which we construct training loss as the mean square difference between predicted and physician-labeled $V_{LV}(t_{ES})$ and $V_{LV}(t_{ED})$.

The network $\widehat{\phi}_{\mathcal{M}}$ is pretrained by generating 3,840 synthetic data points linearly sampled from realistic parameter ranges mapping parameters $\theta_i$ to $V_{LV}(t_{ED})$ and $V_{LV}(t_{ES})$ using a numerical ODE solver (73). We use two fully connected layers trained to minimize mean square error between predicted and outputted $V_{LV}(t_{ED})$ and $V_{LV}(t_{ES})$.

Once predicting $\theta_i$, we can numerically solve for the complete state vector $\mathbf{x}_i$ for each patient. The first state in $\mathbf{x}_i$ is the patient's estimated LV volume $V_{LV}$. The elastance function (Eq. (15)) allows us to obtain LV pressure $P_{LV}(t)$ from volume. We plot the two functions $P_{LV}$, $V_{LV}$ against each other over time to obtain the PV loops. This process is deterministic, as the ODEs/elastance function are fixed by patient parameters $\theta_i$. We also use $V_{LV}(t_{ED})$ and $V_{LV}(t_{ES})$ to compute ejection fraction (EF) as EF $= (V_{LV}(t_{ED}) - V_{LV}(t_{ES}))/V_{LV}(t_{ED})$. We compare parameters and patient PV loops across EF, a routine indicator giving information on cardiac function.

Our model has good correlation between labeled and simulated end-systolic and end-diastolic volumes. Our approach, Med-Real2Sim, achieves a mean absolute error (MAE) of 6.81% for CAMUS and 5.40% for EchoNet in predicting EF (Table 1), comparable to a baseline that uses a 3D-CNN and fully connected layers to directly approximate $\mathcal{F}^{-1}$ using supervised learning without passing the physics-constrained layers $\widehat{\phi}_{\mathcal{M}}$.

| Dataset | MAE (EF) | |
|---|---|---|
| | *Supervised 3DCNN* | *Med-Real2Sim* |
| CAMUS | 6.58 | 6.81 |
| EchoNet | 5.62 | 5.40 |

Table 1: Mean Absolute Error (MAE) for true and predicted EF (in %) by supervised learning and simulated ejection fraction using Med-Real2Sim.

Our model predicts PV loops that vary by labeled EF demonstrating that the model learns variability in patient physical state from our labels (Figure 4(right)). Predicted parameters also vary across patient EF in our model, and the relative variation in high- and low-EF patient groups were replicated across EchoNet and CAMUS (Figure 3). Notably, the relationship between the parameters and patient groups by EF concur with intuited patterns. Increases in $E_{\max}$ and $E_{\min}$ give an increased differential between pressure and volume, which is more plausible at higher EF. $R_M$ and $R_A$ are the circuit resistances associated to the mitral valve and aortic valve, respectively. In the mathematical model, increases in these resistances must either increase pressure or decrease blood flow by Ohm's law. Decreases in either resistance clinical is considered a sign of increased hemodynamic burden, but its diagnostic value for indications including stenosis are controversial and a decrease in resistance is not necessarily predictive of reduce EF (46; 32). In mitral regurgitation, valve resistance decreases, but EF is often increased in the beginning of the disease, which could contribute to the inverse pattern for predicted $R_M$ (52).

## 4.3 Unsupervised disease diagnosis

The PV loops simulated in patient digital twins can serve as indicators for certain diseases that may not be directly labeled in the dataset. Variations in PV loops clinically enables the diagnosis of diverse cardiac abnormalities, which cannot be predicted using EF and LV volumes alone (65). We acquired physician labels for Mitral Stenosis (MS) for a subset of the patients in the EchoNet dataset. These

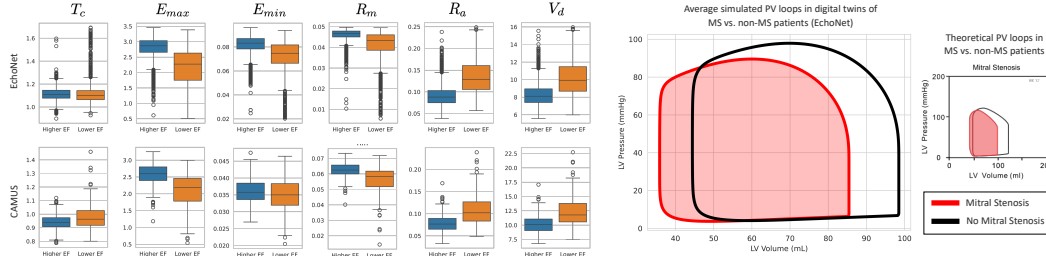

Figure 3: (a) Learned parameters of the digital twin in high- and low-EF patient groups within the EchoNet an CAMUS datasets. (b) Comparison of average (simulated) PV loops in digital twins of non-Mitral Stenosis (MS) patients and MS patients. The plot illustrates differences in simulated hemodynamics in the two groups and agrees with theoretical PV loops for MS patients (48). Depicting of the theoretical PV loop for MS is courtesy of `https://cvphysiology.com/heart-disease/hd009a`.

labels were not included in the training data, and are not correlated with the EF measurement used to train our model (i.e., AUC of EF in predicting MS was 0.5). A total of 263 of 10,024 patients were identified as having MS. We randomly sampled 100 patients with MS and 100 patients without MS from the available dataset of 10,024 patients. We computed an average PV-loop for both the MS and non-MS groups and compared their respective averages to analyze potential differences in Figure 3(b). The model trend captures a distinctive patterns associated with MS: MS reduces LV pre-load and increases pulmonary venous pressures as illustrated in the right plot of Figure 3(b) (3; 48), which aligns with the expected effect of MS on patients' PV loop patterns.

### 4.4 In-silico clinical trials

Using a patient's digital twin, we can simulate their counterfactual PV loops under a hypothetical intervention in-silico to estimate its effect on the patient or to optimize the treatment parameters (e.g., dosage) (56; 72). Here, we demonstrate the feasibility of in-silico trial simulations under baseline and intervention conditions with the patient-specific digital twin $\mathcal{M}(\theta_i)$ (Figures 4(a)). In our setting, the lumped-parameter model of (66) is extended by the inclusion of one new state variable and five new parameters to model the addition of an LVAD. Tuning these new parameters and utilizing our model output allows us to simulate the effect of LVAD on an individual patient in a fully non-invasive manner.

The LVAD intervention has been shown to increase EF in vivo. We demonstrate the same result in-silico using the (66) model (Figure 4). The average EF for the CAMUS and EchoNet populations increase by 17.6% and 18.9%, respectively, with the addition of an LVAD, consistent with reported findings (20; 18)), with patients having lower pre-LVAD ejection fractions experiencing more significant increases. Figures 4(c) illustrates the distributional change of EF before and after LVAD implantation, indicating a significant right shift in the EF distribution following the procedure.

Furthermore, the influence of rotation speed on the PV loop dynamics is a crucial factor in optimizing the therapeutic effects of LVADs. The rotation speed determines the rate at which blood is drawn from the left ventricle and ejected into the systemic circulation, directly affecting the pressure-volume relationship. Studies, such as those by Simaan et al. (66), have explored the dynamic behavior of LVADs under varying conditions, including different initial rotation speeds. It is essential to note that the choice of rotation speed should be tailored to individual patient needs for optimal therapeutic outcomes. The impact of initial rotation speed on the PV loop, such as changes in end-diastolic and end-systolic volumes, should be carefully considered in the calibration process (Figure 4(b)).

### 4.5 Comparison of physics-informed approaches

Our physics-informed pretext task is a computationally efficient generative approximation of the dynamics of the map $\mathcal{M}$. We created a second synthetic dataset of 1000 randomly sampled points to test the out of sample prediction of the physics-constrained layers $\widehat{\phi}_{\mathcal{M}}$ and achieve an MAE of 2.30 on EF. The distribution of the loss is uniform across plausible parameter sets suggesting that our network is not subject to poor approximations in extreme cases (Appendix Figure 9). In contrast, other approaches to learn the forward dynamics of differential equation models include PINNs and

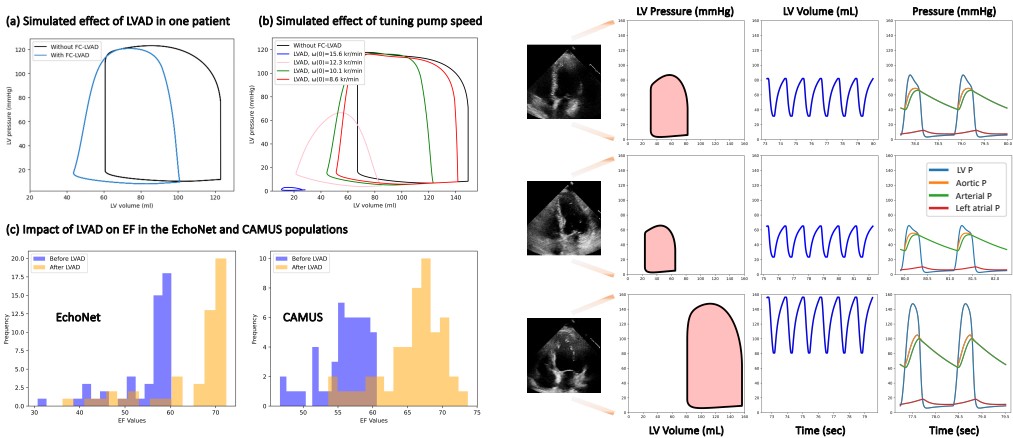

Figure 4: **Counterfactual simulations of the LVAD intervention** (left): (a),(b),(c). **PV loops for patients with normal, high, and low EF** (right).

Neural ODEs. As previously pointed out, these models are trained to learn the dynamics of a single patient, which is incompatible with our goal of learning Eq. (6) with supervised learning.

We compare the ability of our approach to learn the dynamics of the cardiac hemodynamics model for patients to PINN and Neural ODE methods and find it learns just as well on fewer data points (Table 2). Separate PINN and Neural ODE models are trained for each of $N = 50$ sets of patient parameters $\theta_i$. We generated using numerical methods a synthetic time series of 60 points for each of the five states in the cardiac model: $\mathcal{D} = \{\mathbf{x}_i\}_{i=1}^N$. From these synthetic data, we trained both a PINN and Neural ODE approach to predict $\mathbf{x}_i$ for each of the patients computing average MAE on the 20 best results (Table 2). Recall that our method $\phi_\mathcal{M}$ is trained with volume labels (one state) only. We found that only volume labels performed poorly in the Neural ODE settings (Appendix 12). PINNs were implemented in Tensorflow (16). Neural ODEs were implemented in Pytorch using torchdiffeq (11).

|  |  | Architecture | MAE | | Training Time (per Epoch) | | Cost per $\phi_{\mathcal{F}^{-1}}$ Epoch | |
|---|---|---|---|---|---|---|---|---|
|  | *Model* | *Loss* | *Avg. Pt EF* | | *Pt Specific Model (Avg. N=20)* | *Population Model* | *Memory* | *Time* |
| PINN | $\mathbf{x}_i$ | $\lVert f_{\theta_i} \rVert + \lVert \mathbf{x}_i(t) - \hat{\mathbf{x}_i}(t) \rVert$ | 7.67 (N=20) | | 0.16s | NA | $\mathcal{O}(N \cdot L)$ | $\mathcal{O}(N \cdot L)$ |
| Neural ODE | $f_{\theta_i}$ | $\lVert \mathbf{x}_i(t) - \hat{\mathbf{x}_i}(t) \rVert$ | 3.39 (N = 20) | | 6.12s | NA | $\mathcal{O}(N)$ | $\mathcal{O}(N \cdot L)$ |
| $\widehat{\phi}_\mathcal{M}$ | $\mathcal{M}$ | $\lVert \mathbf{x}_{i1}(t) - \hat{\mathbf{x}_{i1}}(t) \rVert$ | 2.30 (N=1000) | | NA | 0.01s | $\mathcal{O}(L)$ | $\mathcal{O}(L)$ |

Table 2: **Comparison of physics informed learning.** We show the Mean Absolute Error (MAE) on ejection fraction (EF). Here $N$ is the number of patient parameter sets and $L$ is the number of layers in the network ($\widehat{\phi}_\mathcal{M}$ or PINN) or the implicit layers i.e. function evaluations of a Neural ODE. Pt = patient.

## 5 Conclusion

We present methodology, Med-Real2Sim, for non-invasive prediction of patient specific physics-based models directly from imaging that can form the basis for further development of medical digital twins. Our experiments showcase the ability of these methods to personalize a high fidelity physics-based model from video data. We demonstrate that information beyond labels is learned in the setting of echocardiography for cardiac pressure-volume loops. Our work may be extended to improve mathematical modeling for medical or non-medical applications. Latent parameters in our experimental setting are clinically informative, and our methodology gives a non-invasive process for estimating their value. Further development of our methods could contribute new non-invasive direct ways of computing such parameters and provide the basis for in-silico digital twin studies.

**Limitations** We highlight that our model experiments are limited in their clinical validation. In using only publicly available datasets with limited labels, we are not able to compare measurements of heart resistance or left-ventricular pressure, which would be necessary to fine-tune such a model for clinical use. While the ability to predict latent states and parameters is the core novelty of our research and

approach, acquiring further data to on these latent states could improve the model substantially. This is a critical future direction for work in building digital twins with our methodology.

## Acknowledgements

The authors would like to thank Ellen Roche, Caglar Ozturk, Yiling Fan, Luca Rosalia, and Yulun Wu for the helpful discussions.

This material is based upon work supported by the National Science Foundation Graduate Research Fellowship under Grant No. DGE 2146752.

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

# A  Mathematical formulation of lumped-parameter hydraulic analogy model

The Simaan et al. (66) model of the cardiovascular system is characterized by the values of the circuit elements, including the capacitances, resistances, and inductance, and by the initial conditions. The model is governed by 17 parameters: eight static parameters that account for the circuit's fixed properties, four parameters that capture the elastance function, and five values describing the initial conditions (see Appendix A.2). The elastance function relates the LV volume to the LV pressure

$$P_{LV}(t) = E(t) \cdot (V_{LV}(t) - V_d), \tag{14}$$

where $V_d$ is the patient specific theoretical volume of the LV at zero pressure. The elastance function is modeled in (66) as

$$E(t) = (E_{MAX} - E_{MIN}) \cdot 1.55 \cdot \left[ \frac{\left(\frac{t_n}{0.7}\right)^{1.9}}{1 + \left(\frac{t_n}{0.7}\right)^{1.9}} \right] \cdot \left[ \frac{1}{1 + \left(\frac{t_n}{1.17}\right)^{21.9}} \right] + E_{MIN}, \tag{15}$$

with $t_n = t/T_{max}$, where $T_{max} = 0.2 + 0.15T_c$, and $T_c$ is the duration of a cardiac cycle. The evolution over time of the state of the circuit at time $t$ is described by a 5-element vector $\mathbf{x}$:

$$\mathbf{x} = [x_1, x_2, x_3, x_4, x_5] = [P_{LV}, P_{LA}, P_A, P_{Ao}, Q_T] \tag{16}$$

where $P_{LV}, P_{LA}, P_A, P_{Ao}$ are pressures of the left-ventricle, left atrium, arteries, and aorta, respectively, and $Q_T$ is total blood flow. We perform a change of variables from the (66) formulation of the model and write

$$\mathbf{x} = [x'_1, x_2, x_3, x_4, x_5] = [V_{LV} - V_d, P_{LA}, P_A, P_{Ao}, Q_T] \tag{17}$$

for ease of computation.

In the model, $t = 0$ corresponds to end-diastole (ED), the end time of the filling phase. We assume the following constraints on the initial states

- There is no restriction on $x'_1$, the initial difference between LV volume and $V_d$. We write $x'_1(0) :=$ start_v.
- For $P_{LA}(0)$, the initial pressure in the left atrium, we know that the filling phase has finished, and the pressures in the left atrium and LV must be equal (63).
- The initial arterial pressure $P_A(0)$, has to be equal to the aortic pressure as at ED there is no current leaving the LV nor the aorta, so the pressures must be equal.
- The initial aortic pressure $P_{Ao}(0)$, has to be larger than the LV pressure at end diastole but by unspecified amount. We have a second free parameter, start_pao that defines this pressure.
- At ED there is not blood leaving the LV, so the initial total flow is $x_5(0) = 0$.

Thus, the initial state vector at ED can be written

$$\mathbf{x}(t = 0) = [\text{start\_v}, \text{start\_v} \cdot E_{MIN}, \text{start\_pao}, \text{start\_pao}, 0] \tag{18}$$

Under these assumptions, the model's degree of freedom is constrained, and the number of independent parameters defining the system becomes 14 (8 for the static elements of the circuit, 4 capturing the elastance function, and 2 for the initial conditions)

$$\Theta = [R_M, R_A, R_C, R_S, L_S, C_A, C_R, C_S, E_{MAX}, E_{MIN}, T_c, V_d, \text{start\_v}, \text{start\_pao}]. \tag{19}$$

The evolution over time of the state vector is derived by application of Kirchoff's current law at five nodes, yielding the non-linear system of equations

$$\dot{\mathbf{x}} = A_1 \mathbf{x} + D_1 p(\mathbf{x}), \tag{20}$$

where $A_1$ is a 5×5 time-independent matrix, and $D_1$ is a $5 \times 2$ time-independent matrix representing the activity of the diodes

$$A_1 = \begin{bmatrix} 0 & 0 & 0 & 0 & 0 \\ 0 & \frac{-1}{R_S C_R} & \frac{1}{R_S C_R} & 0 & 0 \\ 0 & \frac{1}{R_S C_S} & \frac{-1}{R_S C_S} & 0 & \frac{1}{C_S} \\ 0 & 0 & 0 & 0 & \frac{-1}{C_A} \\ 0 & 0 & \frac{-1}{L_S} & \frac{1}{L_S} & \frac{-R_C}{L_S} \end{bmatrix}; \quad D_1 = \begin{bmatrix} 1 & -1 \\ \frac{-1}{C_R} & 0 \\ 0 & 0 \\ 0 & \frac{1}{C_A} \\ 0 & 0 \end{bmatrix}. \tag{21}$$

The vector $p(\mathbf{x})$ is given by

$$p(\mathbf{x}) = \begin{bmatrix} \frac{\max\{x_2 - x_1 \cdot E(t), 0\}}{R_M} \\ \frac{\max\{x_1 \cdot E(t) - x_4, 0\}}{R_A} \end{bmatrix}. \tag{22}$$

The first element of $p$ is the current entering the LV, and the second element of $p$ is the current leaving the LV. These currents depend on state of the valves, which depend on pressure differentials. Equation 20, combined with a set of initial conditions $\mathbf{x}(0)$, provides the evolution over time of the volume of the LV $V_{LV}$ and the pressures in the system. The evolution of $V_{LV}(t)$ and $P_{LV}(t)$ gives patient pressure-volume loops.

## A.1 Addition of a feedback-controlled LVAD

We simulate the addition of a left ventricular assist device (LVAD) to the cardiac system. One end of the device is connected between the left ventricle and the aorta, and the other end is connected between the left ventricle and the left atrium. Many different LVADs have been developed; we take here the case of a rotary-controlled LVAD as described in (12; 66).

The presence of the LVAD induces the addition of a new state variable, $x_6$, representing the blood flow through the device. In addition, the LVAD representation in the electric circuit is described by parameters $R_0$, $R_i$, $\bar{P}$, $\alpha$, $L_i$, $L_0$, $\beta_0$, $\beta_1$, $\beta_2$, — describing compliances, static resistances and a time-varying resistances— and by the pump rotational speed, $\omega(t)$, which is a function of time. The pump speed can be defined beforehand (e.g. a linear increasing function), or it can be updated in real-time based on the state of the system. A more detailed description of the parameters and equations of the LVAD can be found elsewhere (12; 66).

By application of circuit analysis laws, new equations of the circuit describing the cardiovascular-LVAD system are given

$$\dot{\mathbf{y}} = A_2(t)\mathbf{y} + D_2 p(\mathbf{y}) + B(t) \tag{23}$$

for new state vector $\mathbf{y}$, containing the first five variables as in 18 in addition to the blood flow through the LVAD, $x_6(t)$. The new matrices of the system are

$$A_2(t) = \begin{bmatrix} 0 & 0 & 0 & 0 & 0 & -1 \\ 0 & \frac{-1}{R_S C_R} & \frac{1}{R_S C_R} & 0 & 0 & 0 \\ 0 & \frac{1}{R_S C_S} & \frac{-1}{R_S C_S} & 0 & \frac{1}{C_S} & 0 \\ 0 & 0 & 0 & 0 & \frac{-1}{C_A} & \frac{1}{C_A} \\ 0 & 0 & \frac{-1}{L_S} & \frac{1}{L_S} & \frac{-R_C}{L_S} & 0 \\ \frac{-E(t)}{-L_i - L_0 + \beta_1} & 0 & 0 & \frac{1}{-L_i - L_0 + \beta_1} & 0 & \frac{R_i + R_0 + R_k - \beta_0}{-L_i - L_0 + \beta_1} \end{bmatrix} ; D_2 = \begin{bmatrix} 1 & -1 \\ \frac{-1}{C_R} & 0 \\ 0 & 0 \\ 0 & \frac{1}{C_A} \\ 0 & 0 \\ 0 & 0 \end{bmatrix}. \tag{24}$$

$R_k$ is a time-dependent resistance, only allowing current flow through the LVAD when the pressure in the left ventricle is below a threshold noted $\bar{P}$. It is given by

$$R_k = \max\{\alpha \cdot (x_1 \cdot E(t) - \bar{P}), 0\}. \tag{25}$$

The vector $p(\mathbf{x})$ is as defined in 22, and $B(t)$ is given by

$$B(t) = \begin{bmatrix} 0 \\ 0 \\ 0 \\ 0 \\ 0 \\ \frac{-\beta_2}{-L_i - L_0 + \beta_1} \cdot \omega^2(t) \end{bmatrix}. \tag{26}$$

## A.2 Parameters of the lumped-parameter hydraulic analogy model

# B Identifiability of the ODE inverse problem in the cardiac experiments

A unique and valuable attribute of the ordinary differential equation (ODE) in our cardiac hemo-dynamics experiments is its identifiability. In this experiment setting, we can demonstrate that the model $\mathcal{M}$ is well-defined and injective. To show this, we consider the inverse problem posed by finding parameters of the Simaan et al. (66) model, written $\mathcal{M}(\theta) = \mathbf{x}$.

| Parameter | Physiological Meaning | Reference Value | Allowed range |
|---|---|---|---|
| Parameters representing static element of the circuit: | | | |
| $R_M$ | Mitral valve resistance | 0.0050 mmHg·s/ml (66; 83) | [0.005, 0.1] mmHg·s/ml) |
| $R_A$ | Aortic valve resistance | 0.0010 mmHg·s/ml (66; 83) | [0.0001, 0.25] mmHg·s/ml) |
| $R_C$ | Characteristic resistance | 0.0398 mmHg·s/ml (66; 83) | Fixed (0.0398 mmHg·s/ml) |
| $R_S$ | Systemic vascular resistance, related to the level of activity of the patient (23) | 1.0000 mmHg·s/ml (38; 66; 83) | Fixed (1.0 mmHg·s/ml) |
| $C_A$ | Aortic compliance | 0.0800 ml/mmHg (66) | Fixed (0.08 ml/mmHg) |
| $C_S$ | Systemic compliance | 1.3300 ml/mmHg (66; 83) | Fixed (1.33 ml/mmHg) |
| $C_R$ | Left atrial compliance, represents preload and pulmonary circulation (66) | 4.4000 ml/mmHg (66; 83) | Fixed (4.4 ml/mmHg) |
| $L_S$ | Inertance of blood in the aorta | 0.0005 mmHg·s$^2$/ml (66; 23; 83) | Fixed (0.0005 mmHg·s$^2$/ml) |
| Parameters describing the elastance function of left ventricle: | | | |
| $E_{MAX}$ | Maximum elastance, related to the contractility of the heart (67) | 1.5-2.0 mmHg/ml (66), 2.31 (67), 0.7-2.5 mmHg/ml (23) | [0.5, 3.5] mmHg/ml |
| $E_{MIN}$ | Minimum elastance | 0.06 mmHg/ml (67), 0.05 mmHg/ml (66) | [0.02, 0.1] mmHg/ml |
| $V_d$ | Theoretical LV volume at zero pressure | 10 ml (23), 12 ml (66), 20 ml (67) | [4.0, 25.0] ml |
| $T_c$ | Heart cycle duration ($T_c = 60/HR$) | 0.8-1.0 s (66) | [0.4, 1.7] s (i.e. 35-150 beats/min) |
| Parameters describing the initial conditions: | | | |
| start_v | Initial condition related (and not necessarily equal) to the ED LV volume | 140 ml (66) | [0, 280.0] ml |
| start_pao | Initial condition related (and not necessarily equal) to the ED aortic pressure | $\sim$ 77 mmHg (66) | Fixed (75 mmHg) |
| Additional parameters used in describing the addition of the LVAD: | | | |
| $R_o$ | Outlet resistance of cannulae | 0.0677 mmHg.s/ml (66) | Fixed (0.0677 mmHg.s/ml) |
| $R_i$ | Inlet resistance of cannulae | 0.0677 mmHg.s/ml (66) | Fixed (0.0677 mmHg.s/ml) |
| $\alpha$ | LVAD pressure parameter | -3.5s/ml (66) | Fixed (-3.5s/ml) |
| $\bar{P}$ | LVAD weight parameter | 1 mmHg (66) | Fixed (1 mmHg) |
| $L_i$ | Inlet inertance of cannulae | 0.0127 mmHg.s/ml (66) | Fixed (0.0127 mmHg.s/ml) |
| $L_o$ | Outlet inertance of cannulae | 0.0127 mmHg.s/ml (66) | Fixed (0.0127 mmHg.s/ml) |
| $\beta_0$ | LVAD dependent pressure parameter | -0.296 (12) | Fixed (-0.296) |
| $\beta_1$ | LVAD dependent pressure parameter | -0.027 (12) | Fixed (-0.027) |
| $\beta_2$ | LVAD dependent pressure parameter | $9.9025 \times 10^{-7}$ (66) | Fixed ($9.9025 \times 10^{-7}$) |
| $H$ | Circuit pressure difference (inlet-outlet) defining LVAD pressure parameters | $H = \beta_0 x_6 + \beta_1 \frac{dx_6}{dt} + \beta_2 \omega^2$ | |

Table 3: Parameters of the lumped-parameter hydraulic analogy model of the left-ventricle.

First, this function is well defined: given $\theta$, which includes parameters describing the initial system state, there is a unique ODE solution $\mathbf{x}$. Using Picard-Lindelöf, it is sufficient for $A_1\mathbf{x} + D_1 p(\mathbf{x})$ to be continuous in $t$ and Lipschitz continuous in $\mathbf{x}$. As $\mathbf{x}$ describes pressures and flows that cannot be discontinuous in $t$, the first condition is satisfied. As $A_1\mathbf{x} + D_1 p(\mathbf{x})$ is linear in each of the four cases defined by the pairwise possibilities for the rows of $p(\mathbf{x})$ to be either positive or zero, we can take the maximum needed Lipschitz constant amongst these four possibilities to see the entire equation is Lipschitz continuous.

Conversely, given complete knowledge of $\mathbf{x}$, we can find the twelve static model parameters in $\theta$ uniquely. We assume that complete knowledge of $\mathbf{x}$ includes knowledge of the five state variables $(V_{LV} - V_d, P_{LA}, P_A, P_{Ao}, Q_T)$ over time in addition to the complete knowledge of $P_{LV}, V_{LV}$. The

difference between $V_{LV}$ and the first state variable gives $V_d$. The ratio $P_{LV}/V_{LV} - V_d$ gives $E(t)$. The maximum and minimum values of this function give $E_{\max}$ and $E_{\min}$, respectively. The value of $T_c$ is the length of a period of $E(t)$. This gives the four parameters describing elastance.

Applying the Laplace transform $L(f) = \displaystyle\int_0^\infty e^{-st} f(t) dt$ to Equation 20, we have

$$sL(x)(s) - \mathbf{x}(0) = A_1 L(x)(s) + D_1 L(p(\mathbf{x})). \tag{27}$$

The vector $L(p(\mathbf{x}))$ is constant. For any fixed $s$, $L(x)(s)$ is also a constant. Thus, each $s$ gives a linear system of five equations in the eight unknown static parameters governing the Simaan et al. (66) model. Identifying these eight parameters uniquely simply requires choosing enough values of $s$ to solve each row as its own linear system. This shows that $\mathcal{M}$ is injective.

## C  Comparison of problem set up with existent literature

### C.1  Ill-posed inverse problems

Generally, inverse problems are the task of determine inputs $\theta \in \Theta$ from measurements $\mathbf{y} \in \mathbf{Y}$ in the set up $\mathcal{F}(\theta) = \mathbf{y}$ with forward operator $\mathcal{F} : \Theta \to \mathbf{Y}$. Inverse problems are typically ill-posed, meaning that the forward model $\mathcal{F}$ may be non-injective or that small perturbations of measurements $\mathbf{y}$ (i.e. noise) cause large change in the corresponding inputs $\theta$. Often, a variational approach is taken to mitigate these challenges, meaning the problem is reformulated into the task of minimizing over the possible solution set $\Theta$ the sum of a data fidelity term $D$ to represent fitting the model $\mathcal{F}$ in addition to a term incorporating solution criteria $R$ at some level $\alpha$:

$$\min_{\theta \in \Theta} D(\theta, \mathbf{y}) + \alpha R(\theta). \tag{28}$$

For example, $D$ might be $||\mathcal{F}(\theta) - \mathbf{y}||$ to encourage model fitting and $R$ might be $||\theta||$ to encourage input sparsity. This optimization task can been solved using analytical techniques to find $\theta$ directly or using generative learning techniques to approximate $\mathcal{F}^{-1}$ to find $\theta$.

We also seek to solve an ill-posed inverse problem as the forward model $\mathcal{F}$ is non-injective. There are different patient states that could look nearly identical on imaging. In our setting, as we further break down this ill-posed task of inverting $\mathcal{F}$ as that of inverting $\mathcal{F} = \mathcal{K} \circ \mathcal{M}$ for components $\mathcal{M} : \Theta \to \mathbf{X}$ and $\mathcal{K} : \mathbf{X} \to \mathbf{Y}$, we seek to solve an ill-posed inverse problem with partially known forward model. Other strategies for inverse problem with partially known forward model include adversarial methods such as CycleGAN for unpaired signals and measurements and AmbientGAN for training from measurements alone with parametric forward model. CycleGAN simultaneously learns the forward and inverse process by minimizes the expected value of the difference between ground truth inputs and outputs and with forward and inverse model composition outputs (86). AmbientGAN learns the inverse model by discriminating between the distribution of the measurements and distribution of the forward model applied to the modeled inputs (5). For settings with unlabeled data, (19) pioneers Untrained Physically Informed Neural Networks (UPINNs), training a CNN to reconstruct images by reducing the training error between the original measurements and outputs passed through a known forward model.

These existent set ups differ from our task as our partially known model is really the composition of a known and unknown model. We take a *generative* inverse problem approach and develop a model for $\mathcal{F}^{-1}$, which we compare to other generative solutions in Table 4 across the following two taxonomies: known versus unknown forward model $\mathcal{A}$ (row) and paired versus unpaired input-output data (column). This taxonomy mirrors that in (51).

### C.2  Physics informed neural networks and related approaches

Efforts to incorporate knowledge into neural network models to improve training and accuracy include techniques to leverage physical models represented by differential equations (75). In big data settings, physical laws may be learned during modeling. In smaller data settings, incorporating physics into models can improve performance (75). This is especially pertinent to personalized medicine where data is inherently limited. Physics-informed neural networks (PINNs) typically have the goal of either finding solutions (forward) or system parameters (inverse) to systems of differential equations

| | Paired $\{(\mathbf{y}_i, \theta_i)\}$ | Unpaired $\{\mathbf{y}_j\}, \{\theta_i\}$ | Only $\{\theta_i\}$ | Only $\{\mathbf{y}_i\}$ | Paired $\{(\mathbf{y}_i, \mathbf{x}_i)\}$ | No or limited data |
|---|---|---|---|---|---|---|
| Known $\mathcal{F}$ | Reconstruction techniques (34), De-noising autoencoder (49) | Same as training with paired $(\theta, \mathbf{y})$ | Same as training with paired $(\theta, \mathbf{y})$ | UPINN (19) | - | Traditional PINNs for Equation Discovery (60) |
| Unknown $\mathcal{F}$ | Supervised learning (85) | With data distributions: (17), CycleGAN (86) | - | - | - | - |
| Know distribution for $\mathcal{F}$ | - | - | Blind demodulation (28), DeblurGAN (41) | AmbientGAN (5) | - | - |
| Know $\mathcal{M}$ | - | - | - | - | Our method | - |

Table 4: Comparison of problem set up and techniques for ill-posed inverse problems $\mathcal{F}(\theta) = \mathcal{K}(\mathcal{M}(\theta)) = \mathbf{y}$. Abbreviations: UPINN= Untrained Physically Informed Neural Network, GAN=Generative Adversarial Network.

(60; 15). PINN methodologies define training loss as the sum of a data loss and a regularization term, which is the differential equation itself. There is extensive development and mixed success in PINN models for both goals (40).

The set up is the following. Given a state vector $\mathbf{x} \in \mathbf{X}$ dependent on time variable $t$ and differential equation $f_\theta(\mathbf{x}) = 0$ constraining the solutions $\mathbf{x}$ subject to a parametrization $\theta$ with initial or boundary data $D(\theta, \mathbf{x})$, PINNs construct a variational minimization task

$$\min D(\theta, \mathbf{x}) + f_\theta(\mathbf{x}) \tag{29}$$

so that $f_\theta(\mathbf{x})$ acts as a regularizer. This framework can be used to solve both the forward task of solving $f_\theta$ for $\mathbf{x}$ and the inverse task of finding $\theta$. The minimization is over the parameters of the neural network in the case of the forward task and over $\theta \in \Theta$ in the inverse task. We call the forward operator in this set up $\mathcal{M}$, which is the task of solving the differential equation.

Several papers have used PINN approaches to study cardiac function, especially blood flow, including papers that use non-invasive medical imaging data. van Herten et al. (71) learn parameters and solutions of a model for myocardial perfusion using measurements from MRIs. Kissas et al. (39) use a PINN approach to calibrate flow simulation model boundary conditions using data extracted from MRIs. Herrero Martin et al. (29) study cardiac electrophysiology with PINNs and demonstrate the ability of the method to learn a latent variable. Sahli Costabal et al. (62) and Grandits et al. (26) use a PINN approach to study cardiac activation. Our set-up differs in that we learn from images rather than through measurements obtained from images, and we predict informative latent parameters directly and seek a model that works for an entire patient population as opposed to a single patient. Our methodology differs in that the physical model is not integrated into the loss function during training. Instead, we use supervised learning trained on data loss alone to generate a forward model solving the ODE system and transfer this learning.

Drawbacks of the PINN approach include that the differential equation and boundary conditions need not be exactly solved. Other approaches have solved $f_\theta$ as a hard constraint for these same aims in physics-constrained learning (PCL), which mitigates this challenge. A variety of variants on these traditional approaches continue to develop for both forward and inverse problems in various specific problem set ups such as fPINNs (54), SPINNs (61), B-PINNs (82), VPINNs (37), and cPINNs (33), which while not directly relevant in our experiments may be valuable in future digital twin tasks. We compare related techniques to our own with examples in Table C.2.

| | Forward task (learn $\mathbf{x}(t)$) | Inverse task (learn $\theta$) |
|---|---|---|
| Traditional PINNs with $f$ as regularization term | (39) (29) (62) (26) | (71)(39)(29) |
| PCL: $f$ as hard constraint | (84) | (81) |
| PI-NODEs: Train a neural network to learn the non-linear dynamics and a Neural ODE (10) to learn linear components | (42) | (42) |
| P-SSL: $f$ is used to generate synthetic data and use data loss for training a generative solution to inform a larger inverse task | Our method | Our method could be inverted for this task |

Table 5: Comparison of techniques for embedding knowledge of physics models $\mathcal{M}(\theta) = \mathbf{x}(t)$ in neural networks. Abbreviations: PINN=Physics-Informed Neural Network, PCL=Physics Constrained Learning, PI-NODEs=Physics-Informed Neural Ordinary Differential Equations, P-SSL=Physics-Informed Self-Supervised Learning.

### C.3 Neural ODEs

Chen et al. (10) proposed the use of differential equation solvers for solving neural networks upon noticing the similarity between ResNet gradient descent and Euler's method. Viewing neural networks as ODEs means forward propagation is the system solution. It also means that, in contrast, to ResNet architectures, the network does not have a fixed number of layers. Neural ODEs are typically used to model time series data to replace ResNets. The set up is as follows.

Let ODESolver represent a numerical ordinary differential equation (ODE) solver. A fully connected neural network $f$ will represent the ODE the network learns to solve so that a forward pass looks like

$$\text{ODESolver}(f, t_0, x_0, T) \tag{30}$$

for initial condition $\mathbf{x}(t_0) = x_0$ and prediction time series $T$. In this set up, $f$ can represent an ODE that we seek to approximate, analogous to our setting. Neural ODEs are trained by back propagating through the numerical ODE solver. Methods to reduce the computational cost of this backpropagation include the adjoint state variable methods in (10) and augmenting the system to a higher dimensional space (21).

Our fixed pre-training task for leveraging the physical model shares similarities with Neural ODEs in that we train a network to learn to solve an ODE. Our aim differs as a Neural ODE can learn the dynamics for a single patient. The ODE $f$ will be specific to a single patient parameter set $\theta$, but we seek to learn a map for a patient population.

## D  Electric circuit derivation of the (66) model

### D.1  Deriving ODEs from circuit model

The Simaan et al. (66) model has five state variables which are determined by solving a system of ordinary differential equations (ODEs). We derive five ODEs from basic circuit current laws. We obtain equations for the derivative of each of the five state variables. Recall

$$\mathbf{x} = [x_1, x_2, x_3, x_4, x_5] = [P_{LV}, P_{LA}, P_A, P_{Ao}, Q_T] \tag{31}$$

and that in our case, current is flow and voltage is pressure.

The following basic electric circuit facts are sufficient for the derivation.

- For resistors: current = voltage /resistance, i.e. $I = V/R$.
- For capacitors: current = compliance * derivative of voltage with respect to time, i.e. $I = C\frac{dV}{dt}$.

- For inductors: voltage = inductance * derivative of current wrt t, i.e. $V = L\frac{dI}{dt}$.
- Kirchhoff's laws: current must be conserved at any node and voltage sums to zero around any loop.
- $I = \frac{dQ}{dt}$ current is the change in charge $Q$.

The first row of the ODE system is given by conserving the current $I$ at the red node in Figure 5 as

$$-I_3 = I_1 - I_2$$

where $I_3 = (C(t)x_1)' = x_1' + \frac{C'(t)}{C(t)}x_1$, $I_1 = \max(x_2 - x_1, 0)/R_M$ as the voltage flowing across the resistor would be the difference between the voltage at the two capacitors (using Kirchhoff's law) and the diode prevents flow in the negative direction, and $I_2 = \max(x_1 - x_4, 0)/R_A$. Combining and rearranging, we get

$$x_1' = -C'(t)/C(t)x_1 + \max(x_2 - x_1, 0)/(C(t)R_M) - \max(x_1 - x_4, 0)/(C(t)R_A).$$

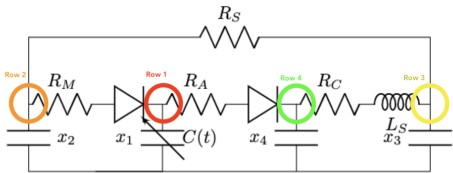

Figure 5: Nodes for deriving circuit equations. The fifth row is derived using total flow and inductance.

The second row is given as the following by conserving current at the yellow circled node in Figure 5

$$(x_3 - x_2)/R_S = C_R x_2' + \max(x_2 - x_1, 0)/R_M$$
$$x_2' = (x_3 - x_2)/(R_S C_R) - \max(x_2 - x_1, 0)/(R_M C_R).$$

The third row is given as

$$C_S x_3' = (x_2 - x_3)/R_S + x_5$$
$$x_3' = (x_2 - x_3)/(R_S C_S) + x_5/C_S$$

by conserving current at the yellow node in Figure 5.

The fourth row is given as

$$C_A x_4' = -x_5 + \max(x_1 - x_4, 0)/R_A$$
$$x_4' = -x_5/C_A + \max(x_1 - x_4, 0)/(R_A C_A)$$

by conserving current at the green node in Figure 5.

The fifth row is given using $L\frac{dI}{dt} = V$ as

$$L_S x_5' = (x_4 - x_3) + \max(x_1 - x_4, 0)/R_A$$
$$x_5' = (x_4 - x_3)/L_S + \max(x_1 - x_4, 0)/(R_A L_S).$$

This relation is written using the relation for inductors above so the left hand of the first line is $L\frac{dI}{dt}$ and the right hand side is voltage. In total, after performing a change of variables of $x_1$ to $x_1' := x_1/E(t)$ these five equations give us Equation 20.

### D.2 Circuit Equivalence

We sought to find a reduced order equivalent circuit to improve model performance but were unsuccessful in finding an identifiable system. We began by ignoring diodes in our circuit to use linear equivalence, as then our circuit would have only linear components (capcitors, inductors, resistors), and we can use the follow result.

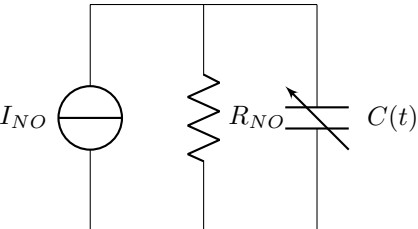

Figure 6: Norton equivalent circuit.

**Theorem D.1** (Norton). *Every linear circuit is equivalent to one with a single current source and single resistor in parallel with a load of interest.*

For our case this gives us the circuit in Figure 6, as we are interested in the capacitor representing pressure change in the left ventricle.

To identify this circuit, we wrote the variables of this new circuit in terms of the state variables in the (66) model. To calculate the resistance of the equivalent circuits, we start by shorting all voltage sources (including capacitors) and opening all inductors. This gives us the circuit in Figure 7.

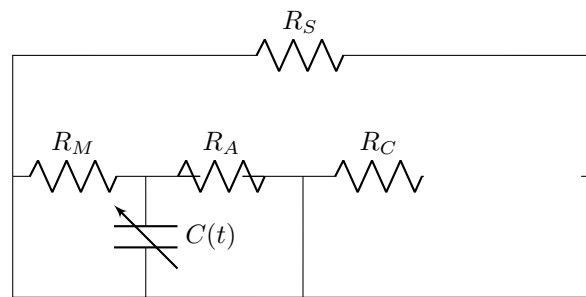

Figure 7: Calculating the resistance of a Norton equivalent circuit.

As this is in fact the resistors $R_S$ and $R_A$ in series and $R_M$ in parallel, using current divider rules, the Norton resistance is

$$R_{NO} = \frac{1}{\frac{1}{R_M} + \frac{1}{R_S + R_A}}. \tag{32}$$

With the addition of diodes, since ours are ideal, when current flow is in the diode's forward direction, we replace the diode with a short. In the reverse direction, we replace the diode with an open, which would vary the calculated resistance to include and exclude resistances.

We would want to leverage the reduced circuit architecture to solve for $P_{LV}(t)$. We know that the current through the load capacitor is $I_{cap} = (C(t)P_{LV}(t))'$ using basic capacitor circuit relations. The Norton equivalent circuit gives a node to solve using Kirchoff's law,

$$I_{R_{NO}} = I_{cap} + I_{NO}. \tag{33}$$

where $I_{R_{NO}}$ is the current flowing through the Norton resistor. Thus, we write

$$I_{R_{NO}}(t) = R_{NO} + (C(t)P_{LV}(t))' \tag{34}$$

$$P'_{LV}(t) = \frac{1}{C(t)}(-C'(t)P_{LV}(t) + I_{R_{NO}}(t) - R_{NO}) \tag{35}$$

giving two unknown state variables $I_{R_{NO}}(t), P_{LV}(t)$ and one ODE, which is insufficient to determine the circuit.

## E   Training details

The EchoNet model, achieving a Mean Absolute Error (MAE) of 5.40%, was trained for 3 hours over 13 epochs with a batch size of 100 and a learning rate of 0.001. In comparison, the CAMUS model,

which reached an MAE of 6.81%, required 2 hours of training over 110 epochs with a batch size of 50 and a learning rate of 0.005.

In our experiments, we conducted a comprehensive hyperparameter search to ensure optimal performance. We employed a grid search approach to systematically explore a range of values for key hyperparameters. Specifically, we evaluated learning rates of 0.001, 0.005, and 0.0005, batch sizes of 50 and 100, and the number of epochs set to 300 and 500. We used the adam optimizer for training.

Models were trained on CPUs, each node featuring 24 cores (12 physical cores with hyper-threading) and 128 GB of RAM.

## F    Additional figures from experiments

### F.1    Validation with Mitral Stenosis labels on EchoNet

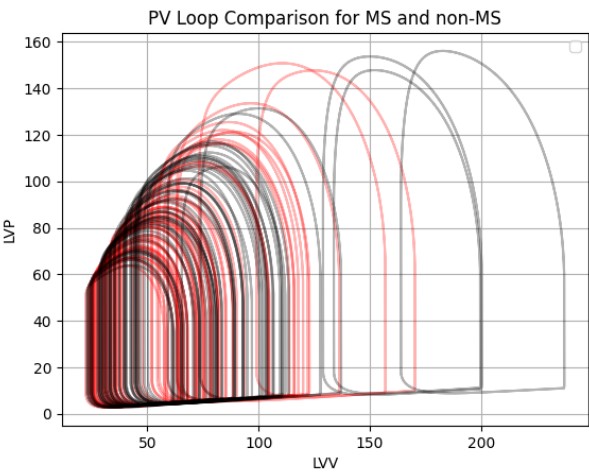

Figure 8: Overlaid PV loops for non-MS patients and MS patients with red patients being labeled as having MS and black patients being without labels. Abbreviations: PV = pressure-volume, MS=mitral stenosis, LVV = Left-ventricular volume, LVP = Left-ventricular pressure.

### F.2    Validation and comparison of physics-informed pretext task

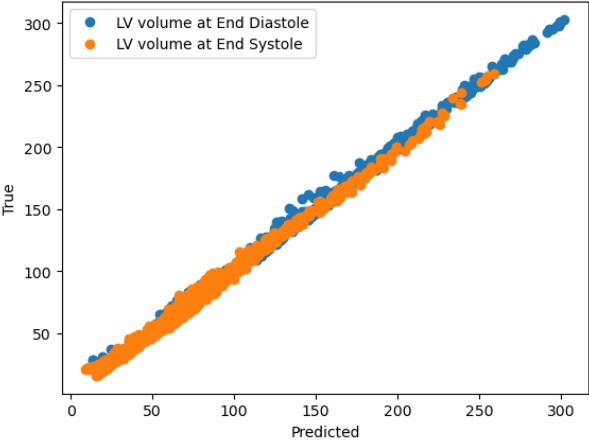

Figure 9: Testing the pre-training task on a second synthetic dataset of 1000 randomly sampled points. The distribution of the loss is uniform across plausible parameter sets suggesting that our network is not subject to poor approximations in extreme cases.

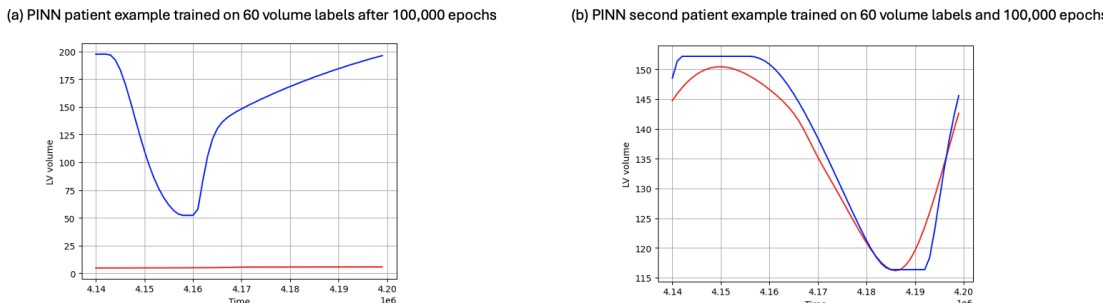

Figure 10: Examples of patient-specific PINNs trained on synthetic data with 60 time points. Some PINNs learn the volume curve well. Others do not learn their patient curve after 100,000 epochs.

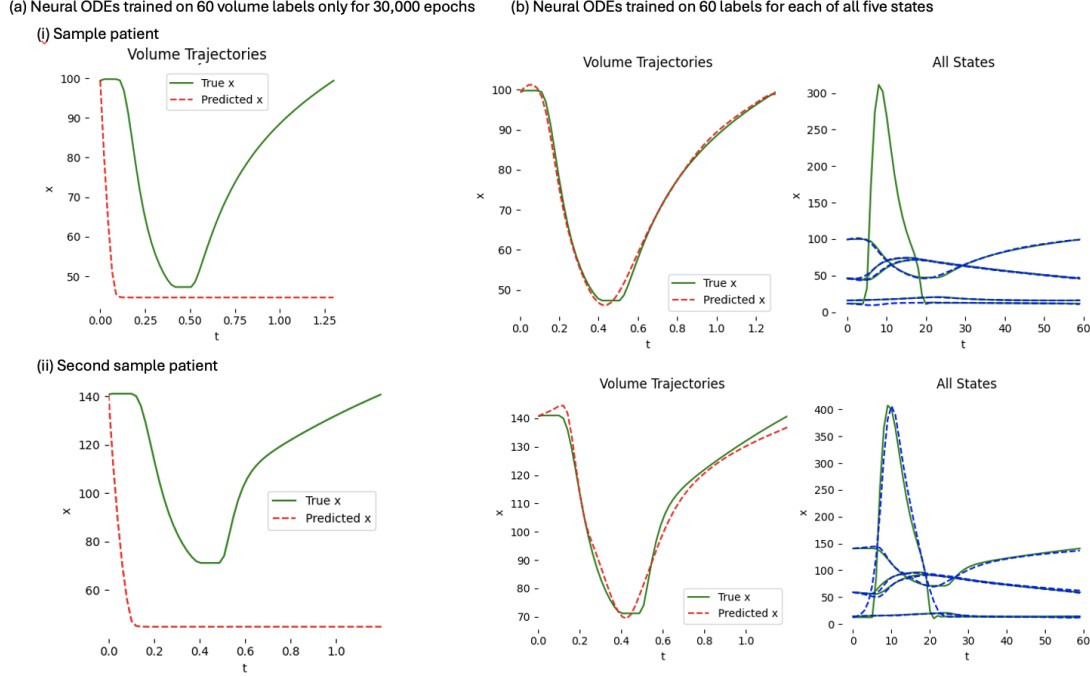

Figure 11: Examples of patient-specific Neural ODEs trained on synthetic data with 60 time points for each of the fives states in the cardiac model or trained only on volume state only.

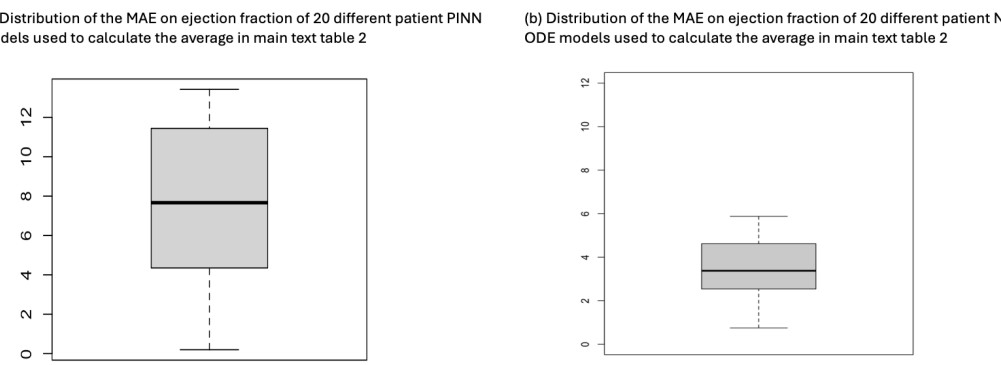

Figure 12: Distribution of MAE on ejection fraction (EF) of the 20 best patient specific PINN and Neural ODE models used to create averages in Table 2.

