# OpenReview forum: "Med-Real2Sim: Non-Invasive Medical Digital Twins using Physics-Informed Self-Supervised Learning"
_NeurIPS.cc/2024/Conference — NeurIPS 2024 poster_

### Official Review · Reviewer_8RJZ · 2024-07-02

**Soundness:** 3
**Presentation:** 3
**Contribution:** 2
**Rating:** 6
**Confidence:** 4

**Summary:**

This paper proposes a novel method for creating patient-specific digital twins using non-invasive patient health data. The authors introduce a physics-informed self-supervised learning (SSL) algorithm that pretrains a neural network on learning a differentiable simulator of the cardiac process. Then, another model is trained to reconstruct physiological measurements from non-invasive data while being constrained by physical equations learned during pretraining. The method is applied to identify digital twins of cardiac hemodynamics using echocardiogram videos, showing good results in unsupervised disease detection and in-silico clinical trials.

**Strengths:**

* The method uses non-invasive data, avoiding time-consuming and complicated patient interventions.
* The model accuracy is enhanced by including a physics-based model during training.
* The authors demonstrate the method's utility in modeling complex physiological processes like cardiac pressure-volume loops with open-sourced datasets.

**Weaknesses:**

* Simplifications/assumptions in the Windkessel and LVAD models might not fully capture the complexity of the heart dynamics.
* The results might be sensitive to low quality non-intrusive data, which can affect the global accuracy of the method.
* It is not clear the demographic diversity of the echocardiography dataset. Thus, the model might not generalize well across all the segments of the population.

**Questions:**

* Lines 251-255, Table 3: Which criteria was used to select the learnable and the fixed parameters of the model? Are these selected by using some kind of sensitivity analysis based on the state-space matrices in Eqs. 21 and 24?
* Line 708: Appendix C1 already addresses the ill-possedness of the inverse problem. That means that the trained model could potentially assign the same digital twin to two different patients due to the similarity of their echocardiograms. Have the authors found any difficulty in this regard?
* Have the authors considered using an easier and more accessible nonintrusive techniques such as electrocardiograms? Would the use of several modalities of non-intrusive data for the same patient increase the performance of the model?

Minor comments:
* Line 227: "tune-able" might refer to "tunable".
* Line 252: Incorrect reference, Table A.2. might refer to Table 3.
* Line 814: Incorrect reference, Figure D.2. might refer to Figure 7.
* Figure 7: The figure model seems to be incomplete/offset.
* Equation 35: A parenthesis is missing after $R_{NO}$.

Final comment: The paper is well structured, the results are promising and the methodology has moderate impact on the field. However, it's still unclear if this model could handle the variability and complexity of human physiology by only non-invasive measurements. Based on the comments above, I reccomend a weak accept.

**Limitations:**

Limitations were addressed appropriately.

---

> ### Author Rebuttal · Authors · 2024-08-07
>
> We thank the reviewer for the very thoughtful comments and feedback!
>
> **Weaknesses:**
>
> Thank you for highlighting these issues. Please see below a point-by-point response to your concerns.
>
> * While the Windkessel model does not fully capture the complexity of cardiac dynamics, this simplicity is actually a strength of our approach. The heart is an incredibly complex organ system, with millions of parameters describing its electrophysiology, hemodynamics, and biomechanics. However, it is not always necessary to simulate the entire cardiovascular system to make specific diagnostic or treatment decisions. Our model focuses on predicting pressure-volume (PV) loops relevant to diagnostic and treatment procedures for end-stage heart disease. In this context, the Windkessel model is the most parsimonious in-silico simulator for the relevant physiological processes. Additionally, we want to stress that our main contribution is a general two-stage physics-informed self-supervised learning approach and not a specific in-silico model—this approach is general enough to incorporate any cardiovascular system simulator, including more complex lumped circuit models of the arterial system.
> * Please note that non-invasive data is not necessarily of low quality; it simply means the data was acquired without invasive procedures, such as catheterization. Echocardiograms, which we use in this paper, are a widely-used, gold-standard non-invasive method for diagnosing cardiac function. Our paper proposes a new method that extracts additional valuable information from echocardiograms, which previously required invasive procedures and were impractical in most primary care settings. This method enables new use cases and does not compromise the accuracy of any existing procedures.
> * This dataset includes a reasonably diverse demographic, comprising 68% White, 14% Black, 7% Asian patients, and individuals from other ethnicities. Additionally, 55% of the patients are male. We will provide a detailed demographic performance breakdown in the Appendix of the final paper.
>
> **Responses to questions:**
>
> **Question 1:** We selected learnable versus fixed parameters based on the intended use case of predicting cardiac left ventricular pressure-volume (PV) loops in combination with a sensitivity analysis to understand how they affect the model. Ultimately, parameters directly related to the circuit analog of blood flow in the left ventricle were allowed to be variable, while others were fixed to isolate the prediction of these dynamics. We first defined intervals of validity for each of the 14 independent parameters, based on how variations of each parameter affects the PV loops (realistic shape and magnitude of pressures of volumes), while ensuring that the volume ranges and EF ranges of the two echo datasets were covered. These were centered around reference values from the literature. For the 14 independent parameters, we did perform a sensitivity analysis based on the state-space matrices in Eqs. 21. We selected the final 7 learnable parameters, based on how changes in these parameters affect the volume space, their physical meanings, their resulting PV loop shapes, and their ability to encompass the volume ranges of the two echo datasets. We **include a table in the global rebuttal PDF that summarizes the sensitivity analysis results and modeling choices**.
>
> **Question 2:** It is definitely true that, in general, patients with similar echocardiograms may be assigned identical digital twins and it is also true that this may not capture sufficient differences between patients. This is a general issue encountered in all inverse modeling problems, and is not peculiar to our setup. In our setting, we use a lumped parameter model with a small set of parameters that we have reasons to believe will be able to be distinguished between patients because they have a visual representation in an echocardiogram. This means that there are correlations between visual features and values of the in-silico model parameters that allow us to uncover meaningful differences in the digital twins that can distinguish, for example, mitral stenosis (Fig. 3(b)). We believe that studying the identifiability of specific models, developing general conditions for identifiability of physics-based models and developing new learning procedures that improve the chances of model identification are all very interesting directions for theoretical research that complements our applied method. One potential direction to improve identification would be incorporating patient level meta-data or other modalities as is done in other digital twin settings to enhance the model’s ability to distinguish between patients to make a measurement model more nearly injective.
>
> **Question 3:** It would be really interesting to study the use of electrocardiograms in this setting and even more interesting to combine these data assets. Echocardiograms were the natural choice for our model because they are a standard tool for encoding information about left ventricular pressure in addition to left ventricular volume, which they are routinely used to assess [Popescu et al 2022]. However, we recognize that incorporating additional non-invasive modalities such as electrocardiograms could potentially enhance the model’s performance. We are curious to explore the possibility in future studies.
>
> Thank you for catching the references and minor typos, we will happily update those. Figure 7 is intended to show shorting the circuit to calculate an equivalent circuit.
>
> _Bogdan A Popescu, Carmen C Beladan, Sherif F Nagueh, Otto A Smiseth, How to assess left ventricular filling pressures by echocardiography in clinical practice, European Heart Journal - Cardiovascular Imaging, Volume 23, Issue 9, September 2022, Pages 1127–1129, https://doi.org/10.1093/ehjci/jeac123_

---

> > ### Comment · Reviewer_8RJZ · 2024-08-12
> >
> > I thank the authors for the rebuttal, I have no more concerns.

---

### Official Review · Reviewer_CAet · 2024-07-12

**Soundness:** 3
**Presentation:** 3
**Contribution:** 3
**Rating:** 6
**Confidence:** 4

**Summary:**

This paper introduces a novel methodology for identifying patient-specific digital twins using noninvasive medical imaging, particularly focusing on cardiac hemodynamics. By leveraging a physics-informed self-supervised learning approach, the research addresses the challenge of modeling digital twins without invasive data collection. The process involves pretraining a neural network to simulate physiological processes and then fine-tuning it to reconstruct physiological measurements from noninvasive modalities, constrained by the learned physical equations. This framework allows for the simulation of patient-specific physiological parameters and the potential for conducting in-silico clinical trials and unsupervised disease detection.

**Strengths:**

1. The paper introduces a cutting-edge method combining physics-informed neural networks with self-supervised learning to tackle the inverse problem of estimating physiological parameters from noninvasive imaging data.

2. By utilizing noninvasive data, the proposed method significantly reduces the need for invasive procedures, enhancing patient safety and comfort, and potentially broadening the applicability of digital twin technology in routine clinical practice.

3. The methodology's ability to simulate detailed physiological conditions and interventions opens up vast possibilities for its application in personalized medicine, including unsupervised disease detection and in-silico clinical trials, which can significantly accelerate the development of therapeutic strategies.

**Weaknesses:**

1. Insufficient performance comparison. The paper only compares PINN and Neural ODE methods. There are many variations of PINNs methods which outperform the original one. The paper should choose more solid baselines.

2. No ablation study. The paper proposed a two-stage training strategy, but it didn't show why it is necessary.

**Questions:**

1. How do the results of the model change if the estimated parameters are dynamic rather than constant? Considering the medical context where a patient's condition is continually changing, assuming constant parameters seems unrealistic. How might this affect the reliability and accuracy of the model used in such scenarios?

2. Can you explain why blood flow is often modeled using an electrical circuit analogy, and why it adheres to Kirchhoff's laws of voltage and current?

**Limitations:**

The proposed method requires a pretty strong assumption of the PDEs dynamics. I am a little bit worried about the training efficiency of this method. Can it be used as a real diagnostic tool? Additionally, I am intrigued by the potential of this method to be applicable across a broader range of medical disciplines.

---

> ### Author Rebuttal · Authors · 2024-08-07
>
> We thank the reviewer for your very thoughtful comments! Below, we provide a detailed response to the weaknesses and questions.
>
> **Weaknesses:**
>
> Please see below a point-by-point response to your concerns.
>
> * The limited baseline comparisons stem from the novelty of our problem setup. While there are many variants of Physics-Informed Neural Networks (PINNs) in the literature, not all are suitable for our specific problem. Our setup is unique because it involves a partially known model that combines both known and unknown components. As explained in lines 721-725, this setup can naturally be approached using a generative inverse problem approach to model $\mathcal{F}^{-1}$. We compare this model to other generative and PINN solutions in Table 4, using two taxonomies: the known versus unknown forward model (rows) and paired versus unpaired input-output data (columns). Table 4 highlights that the various PINN variants make different assumptions about the problem setup, making them unsuitable as baselines for our case. Therefore, we chose to compare our approach only with baseline PINN and Neural ODE methods. Our goal was to demonstrate that our surrogate model achieves a different goal but still has a comparable understanding of the underlying physics as these standard approaches, rather than showing superior performance in learning physics. We are open to implementing additional baselines if the reviewer suggests specific PINN variants that are applicable to our setup.
> * We did conduct an ablation by removing the physics-based model to evaluate its impact on ejection fraction (EF) predictions (Table 1). Additionally, we explored other feature extraction architectures, such as LSTM and transformer models, but these did not achieve the same level of loss minimization as our 3D-CNN approach. For the two-stage training strategy, we also experimented with alternative differentiable ODE solvers. However, these solvers were not compatible with our architecture and resulted in similar parameters across different patients or introduced additional numerical error. Consequently, we opted for our two-stage approach, which includes a physics-informed pretext task and physics-guided fine-tuning with 3D-CNN. We are very open to suggestions from the reviewer for additional ablation studies that could further validate our approach and their implications for our set up.
>
> **Responses to questions:**
>
> **Question 1:** First, we want to clarify the difference between the patient's cardiac states and cardiovascular parameters. Our model is dynamic: it analyzes an ultrasound video to extract a cardiac state that varies over time, characterized by changes in pressure and volume throughout the video, which lasts for several minutes. The underlying parameters, however, are fixed because they depend on the heart's anatomy, deformities, and tissue properties, which change over days, months or years, not within the short duration of a single echocardiogram acquisition. While modeling the long-term evolution of cardiac health is a fascinating problem, our model can still address this setup by applying our two-stage procedure to new echocardiograms acquired over different years.
>
> **Question 2:** The analogy between cardiac blood flow and electric circuits is a classical result in hydrology and fluid mechanics. In this analogy, blood flow is analogous to electrical current as it is driven by pressure differences and encounters resistance like electricity is driven by voltage and material resistance. Valves act as diodes, capacitors emulate the elastance behavior of heart chambers, and heart tissues induce resistance analogous to that in electric circuits. This analogy is often referred to as the “drain-pipe theory” and is widely used in fluid mechanics beyond cardiovascular modeling. The class of models studied in our paper, called Windkessel models, with varying complexity have been used to model cardiac dynamics for many years, though other classes of models do exist. The model we take to be fixed for our paper was validated in the Simaan et al paper (which we specifically used for its incorporation of an LVAD). The authors synthetically perturbed the model to demonstrate the behavior was as clinically expected and compared to real human measurements to demonstrate accuracy.
>
> Kirchhoff's laws are adhered to in the model simply because they are adhered to in any electric circuit. In this way, Kirchhoff's Voltage Law parallels the conservation of energy in the circulatory system, where the sum of pressure drops around a closed loop equals the pressure driving the flow. Kirchhoff's Current Law reflects the conservation of mass in blood flow, ensuring that the total blood flow entering a junction equals the total flow exiting.
>
> **Finally, we want to address your overall comment** and clarify that we do not make strong assumptions about PDE dynamics. Instead, we use a low-fidelity model that exactly characterizes cardiac hemodynamics in the heart chambers relevant to cardiovascular diseases, while ignoring the details of blood flow in other arterial peripherals. We strongly believe that our approach has broad applications for individualized treatment of end-stage heart failure. This includes not only the use of LVAD, as demonstrated in Section 4.4, but also any other medical device with a corresponding in silico model. Additionally, our method can be used to diagnose diseases not typically identified using echocardiograms alone, such as mitral stenosis.

---

> > ### Comment · Reviewer_CAet · 2024-08-13
> >
> > Thank you very much! Your reply almost addresses my concerns.

---

### Official Review · Reviewer_32qb · 2024-07-14

**Soundness:** 3
**Presentation:** 3
**Contribution:** 2
**Rating:** 5
**Confidence:** 4

**Summary:**

I have read this manuscript during ICML review. It looks the same so I copied my previous review.

The authors presented a method to infer the physical parameters θ of physiological process (heart pumping blood) from noninvasive observation y (the echo image). The mapping from y to θ cannot be directly learned due to the lack of paired data. Instead, they find that an intermediate variable x_bar (the EF) can be annotated by experts (x_bar=g(y)) which can also be calculated based on θ (x_bar=m(M(θ))). Thus the observable pair (y, x_bar) provides the supervision for learning θ = F_inv(y), through the relation x_bar=m(M(F_inv(y))) where m is rule-based and M is a solution to ODE. They first train a surrogate network to approximate M(θ) through synthetic simulation, followed by learning the parameter of F_inv on observations {(y, x_bar)}.

**Strengths:**

1. The paper is clearly written and easy to read with clear symbols.
2. The idea of introducing a physical model helps better characterise the physiological system and provide a learning method.
3. The surrogate model overrides the cost of solving ODE and make it differentiable during learning.
4. The inference of physical parameters helps to perform virtual experiments.

**Weaknesses:**

1.The authors compared their methods in predicting EF from echocardiogram with supervised 3DCNN but did not outperform them in MAE. It should be noted that the EF is calculated on ED and ES segmentation and supervised segmentation network is expected to perform even better.
2.Lack of validation. The authors performed validation by comparing EF derived from the physical parameters to the observed EF. But this does not guarantee the correctness of their physical parameters. In fact, arbitrary intermediate variables can be defined and could also lead to the comparable prediction of EF.
3.Training the surrogate model of x=M(θ) need to generated synthetic samples while this is domain-dependant. Whether the learned mapping fits extreme or shifted situations is unknown, lacking of uncertainty analsysis.

**Questions:**

1.Could the authors provide more solid validation of their inferred physical parameters?
2.Could the authors provide the uncertainty/robustness/generalisation ability of the surrogate model?

**Limitations:**

My biggest concern is that the method lacks ground-truth validation, at least some samples.

---

> ### Author Rebuttal · Authors · 2024-08-07
>
> Thank you for your review of our paper. We greatly appreciate your and other reviewers feedback from our ICML submission, and we have implemented changes to address these comments in our new submission. We are happy to share that we have **added additional validation and comparison of the physics based surrogate model in response to your feedback**. Specifically, we compared this approach to a baseline PINN approach and a Neural ODE approach, which demonstrate similar performance but fundamental differences in set up (Table 2). We also took into account your suggestion of further validating the surrogate model and its generalizability by adding predictions on an additional out-of-sample synthetic dataset in Section 4.5 and Appendix Figure 9.
>
> In response to your specific concerns, we restate or add the following in a point-by-point response.
>
> **Weakness 1:** Thank you for your observation regarding the comparison with supervised 3DCNN and other segmentation models in predicting EF from echocardiogram data. While we acknowledge the potential for supervised segmentation models to yield superior results in terms of mean absolute error (MAE) for EF prediction, it's important to clarify that our study's primary objective differs from that of these supervised algorithms. In our research, the primary focus is utilizing the volume states that compose EF (volume at ES and ED) as a clinically significant label to facilitate the training of our model for latent parameters and PV loops. We incorporate physics as fixed layers for this purpose, a process which may marginally compromise the model's EF prediction accuracy. In Table 1, we showcase that the sacrifice in EF prediction accuracy is minimal and acceptable within the context of our objectives. We acknowledge that certain state-of-the-art Unet models designed for segmentation tasks may indeed achieve superior MAE in EF prediction. However, it's crucial to note that these models are specifically optimized for segmentation purposes, whereas our methodology integrates physics-informed learning to achieve broader objectives beyond segmentation alone.
>
> **Weakness 2:** Thank you for bringing up the limitation on the validation of our latent parameters. Evaluating solely comparing EF derived from physical parameters to observed EF for validation is clearly insufficient as you point out. Validating these individual parameters is of course challenging given that they are parameters of a physical model rather than exact patient measurements. In our particular experimental setting, our parameters certainly have physiological meaning, which we describe qualitatively, but they are not directly measurable. For example, Rm, mitral valve resistance, in our model is an electric circuit analogy parameter that is not equal to measures of mitral valve resistance in clinical practice. We sought to perform qualitative validation, i.e. describing qualitative relationships between Rm and actual mitral valve resistance and in using predicted PV loops to predict mitral valve stenosis disease labels. In future work, we would seek to find correlations between model parameters and physiological effects or conditions. For example, [Lamberti et al 2024 below] show correlation between pulmonary vascular compliance, a parameter similar to those in our model, and right heart decompensation with LVAD implantation.
>
> _Kimberly K. Lamberti et al, Dynamic load modulation predicts right heart tolerance of left ventricular cardiovascular assist in a porcine model of cardiogenic shock. (2024).DOI:10.1126/scitranslmed.adk4266_
>
> **Weakness 3:** The surrogate model/pretext task is definitely intended to be domain dependent and would need to be trained on synthetic data for any new in silico mathematical model. The data will always be synthetic for pretraining the physical dynamics, so it would not be dependent on the image data’s collection. In our case, the surrogate model is trained on a synthetic multidimensional grid filling a space of the form [a_1, b_1] x .. x [a_n, b_n], where [a_i, b_i] is the validity interval for the parameter θ_i, and θ=[θ_1,...,θ_n] and is chosen accordingly to the problem addressed. From this synthetic dataset, covering all possible states (and thus all possible patients), the surrogate model M(θ) is domain independent, and capable of predicting M(θ) even for extreme or shifted situations (since these are included in the synthetic dataset). We add an out-of-sample prediction on 1000 new points and show a uniform distribution of errors and comparison of this method to other physics informed approaches to demonstrate that it learns just as well and is better suited to our set up.

---

> > ### Comment · Reviewer_32qb · 2024-08-11
> >
> > Thank you for your rebuttal. My concerns have been addressed to some degree. I have increased my rating.

---

### Official Review · Reviewer_LjNF · 2024-07-19

**Soundness:** 1
**Presentation:** 3
**Contribution:** 2
**Rating:** 3
**Confidence:** 3

**Summary:**

The paper proposes a method to identify parameters for digital twin models of patients using non-invasive health data, eliminating the need for invasive procedures. This method focuses on scenarios like cardiac hemodynamics, where traditionally invasive measurements (e.g., through catheterization) can be predicted using non-invasive data (e.g., echocardiograms). The novelty of the method is to solve the associated inverse problem specifically for that patient, so that personalized predictions can be performed.

The proposed method uses a two-step SSL approach that structurally resembles pretraining and finetuning in SSL. First,  a neural network is pretrained on synthetic data to learn the forward dynamics of a physics-based model. Then the pretrained model is then used to train another network on actual non-invasive patient data to predict physical parameters.
Application to Cardiac Hemodynamics:

The paper illustrates how to apply the above method for cardiac hemodynamics using echocardiography. This allows the prediction of patient-specific pressure-volume (PV) loops using non-invasive echocardiogram videos.

**Strengths:**

The paper is overall well-written and it identifies a real problem with potential high impact: the design of personalised medical twins to avoid invasive procedures

**Weaknesses:**

The authors focus only on a very specific medical use-case, they don't try to generalize their method to more cases. In the introduction the authors illustrate this as a generic approach, therefore I was surprised to not find an attempt to support their vision with more application examples. Without the demonstration that this method can have a broader use I don't think this paper is suitable for presentation at this conference. Also, the authors don't run convincing ablations to demonstrate that their approach is sounding.

**Questions:**

Have you experimented with this method beyond Cardiovascular Hemodynamics?
You implement a 3D-CNN - how did you arrive at such architecture? have you run ablations?

**Limitations:**

The authors mention the limited clinical validation. This is certainly one limitation to take in account. I would consider also the lack of generalization to different clinical procedures.

---

> ### Author Rebuttal · Authors · 2024-08-07
>
> Thank you for the feedback on our paper!
>
> The proposed method is indeed general and not limited to the cardiovascular system example. Any physical or biological system that can be described through ordinary differential equations can utilize our approach. The problem setup in Section 2.1 and the training/loss functions in Section 2.2 are designed to be very general, capturing any mapping from non-invasive observations to a physical system defined by differential equations. We chose to focus on cardiovascular hemodynamics as our example because our contribution is conceptual rather than empirical. By studying a specific disease area in depth, we were able to illustrate a wide range of use cases for digital twins, from diagnosis to simulation of counterfactual treatment outcomes. These use cases were previously not possible, and our goal was to demonstrate the potential rather than validate a new model for an existing use case at scale. We believe that our in-depth focus on a disease area enhances rather than undermines our contribution.
>
> To give another example for a problem setup in oncology where the exact same framework can be applicable, consider the in-silico model proposed in [Baldock et al 2013], which describes the use of an ordinary differential equation for modeling proliferation and invasion of tumor cells. The two parameters of this model: dispersion D and proliferation p, have physiological meaning and impact on cancer prognosis [Baldock et al 2013]. They have been proposed to be measured using a mathematical formula from diffusion weighted MRI [Ellingson et al 2010 below]. Our approach could learn the dynamics of the ODE and build a non-invasive direct approach for these parameters to be extracted from images with a 3DCNN, provided data is available. By demonstrating a method for creating digital twins that can be generalized, our work could encourage more public sharing of relevant healthcare datasets, fostering the development of new digital twins across various medical fields. This is a crucial step towards broader applications and greater impact in medical research.
>
> In response to your questions regarding the choice of architecture and ablation studies, we want to stress that our framework is not specific to this architecture. We found the 3D-CNN architecture to be the most empirically performant architecture in predicting ejection fraction using the PINN layers compared to other architectures we tried, such as LSTM and transformer models. This is in line with previous results that showed the effectiveness of 3D-CNNs in modeling echocardiography [Ouyang et al 2021]. Regarding ablation studies, we conducted an ablation by removing the physics-based model to assess its impact in EF predictions. **We are very open to suggestions from the reviewer for additional specific ablation studies** that could further validate our approach and their implications for our set up.
>
> _Ouyang, D., He, B., Ghorbani, A., Yuan, N., Ebinger, J., Langlotz, C.P., Heidenreich, P.A., Harrington, R.A., Liang, D.H., Ashley, E.A. and Zou, J.Y., 2020. Video-based AI for beat-to-beat assessment of cardiac function. Nature, 580(7802), pp.252-256._
>
> _Ellingson, B.M., LaViolette, P.S., Rand, S.D., Malkin, M.G., Connelly, J.M., Mueller, W.M., Prost, R.W. and Schmainda, K.M. (2011), Spatially quantifying microscopic tumor invasion and proliferation using a voxel-wise solution to a glioma growth model and serial diffusion MRI. Magn. Reson. Med., 65: 1131-1143. https://doi.org/10.1002/mrm.22688_

---

> > ### Comment · Reviewer_LjNF · 2024-08-13
> >
> > I have read the authors reply and all the other reviews. I appreciate the effort of the authors in providing more details and explanation, they helped me understand the paper and their motivations better. I slightly increased my score. However my concern on the generalizability of the method and the missed opportunity to demonstrate it remains. (and I can see it's a shared concern among the reviewers).

---

### Author Rebuttal · Authors · 2024-08-07

We thank all reviewers for their valuable feedback. We would like to take this opportunity to summarize the key contributions of our work, address common concerns across the reviews, and clarify some aspects of our methodology that may have not been fully appreciated.

**Summary of Contributions**
Our paper introduces methodology for tuning patient-specific parameters of medical digital twins with non-invasive data. We do so by creating a framework for solving a composite inverse problem to reconstruct physiological states from indirect measurement non-invasively. We build a physics-informed self-supervised learning (SSL) algorithm that first pre-trains a neural network to learn a differentiable simulator of the mathematics of the digital twin model which is then used to constrain the reconstruction of physiological measurements from data. This enables diverse applications from simulations of patient-specific models, which we demonstrate in our experiments in the setting of cardiac hemodynamics and the facilitation of in silico trials (for left-ventricular assistance devices, LVADs) and disease detection (mitral stenosis).


**Key Strengths**

* **Innovative methodology:** Our method is designed to be suitable to a challenging real world inverse problem. We combine the strengths of physics-informed models with self-supervised learning to address the challenge of modeling personalized digital twins without direct measurement.
* **Non-invasive data:** Our approach is built with non-invasive indirect measurements. This has the potential to spur research into the development of medical digital twins suitable in a more diverse patient population.
* **Broad application:** The proposed framework allows for the simulation of detailed physiological parameters and interventions in any set up where a process can be modeled using ordinary differential equations. This generalizability will allow for further research for more widely applicable digital twins of various health systems and non-invasive modalities.

**Common Themes:**

* **Architecture and ablation studies:** We performed ablation studies to evaluate the impact of removing the physics-based model component and explored alternative architectures. The results showed that removing the physics-based model still allowed for comparable EF predictions, indicating that our model’s supervised learning capabilities were not compromised while enabling digital twins simulation. We explored various video feature extraction frameworks including LSTM and transformers besides 3DCNN and alternative differentiable ODE solvers finding that none performed as well in our experimental setting. We are open to suggestions to the reviewers of additional ablation studies to further validate our approach and set up but want to stress that our framework is not specific to any particular architecture.
* **Generalizability of approach:** Our manuscript presents the example of the application of our methodology for modeling cardiovascular hemodynamics with non-invasive echocardiogram data. For this experiment, we selected parameters in a simplified dynamic lumped parameter model with the specific focus of evaluating patient cardiac pressure-volume loops. We evaluated the performance on two public echocardiogram datasets with diverse demographics and studied this experiment in depth to demonstrate a wide range of potential use cases for models developed in this framework. However, the problem set up and training are designed to be very general so that they might be used to develop digital twins for any system that can be modeled with ordinary differential equations using a mapping from a set of non-invasive observations. Our method is conceptual, aiming to foster data sharing and encourage broader applications in medical research.
* **Performance comparison and baselines:** We selected comparisons, specifically Physics-Informed Neural Networks (PINNs) and Neural ODE methods, to emphasize the fundamental differences in setup and to demonstrate differences in our surrogate model despite a comparable understanding of the underlying physics as state of the art methods. There are many variants of PINNs in the literature but not all are suitable to our set up. Our method is designed to approximate known and unknown components of an inverse problem. We propose a generative inverse problem approach that will work across parameter sets, which contrasts with the PINN and Neural ODE approaches that focus on learning physics for individual patients. Despite potential advantages of advanced PINN variants, our approach is tailored for capturing broader inverse relationships rather than optimizing for single-patient physics learning. We compared our approach to these baselines to illustrate similar performance while highlighting our method’s unique setup and goals. We remain open to incorporating additional relevant baselines if they align with our specific framework.

---

### Decision · Program_Chairs · 2024-09-25

**Decision:**

Accept (poster)

**Comment:**

This manuscript proposes a novel method for creating patient-specific digital twins focused on cardiac hemodynamics. The general idea is how to infer from a non-invasive measure an invasive measure. The domain specific task aims to predict the physiological signal such as heart pumping blood from the recording of an echocardiogram image. The novelty of the method is to solve the associated inverse problem specifically for that patient, so that personalized predictions can be performed. The proposed method uses a two-steps Self Supervised Learning (SSL) approach that can be summarized by a step of pre-training and a step of fine tuning. A neural network is pretrained to learn from the data of a differentiable simulator the parameters of cardiac process. Then, another model is trained to reconstruct physiological measurements from non-invasive data while being constrained by physical equations learned during pretraining.

The rebuttal allowed the clarification of several aspect and the Reviewers agreed that the discussion was fruitful for a better understanding of the proposed method. No major concerns affect the soundness of this work after the rebuttal.

The controversial aspect of this work remains how should be perceived from the scientific community. The Authors believe that their contribution is general and the application to cardiac hemodynamics has only the purpose of an explanatory example. The Reviewers complain about this claim at least for two reasons: (i) the assumptions for the generalization are not clearly formulated, (ii) the successful application of this approach to other use cases is purely speculative and without empirical evidence.
The proposed framework assumes at least a couple of premises: a simulator that encodes the relationship between a target variable x and an intermediate variable w, a collection of data where the annotation between an input variable y and the variable w is available (where x would be the invasive measure and y the non invasive measure). Even though this pattern can be recognized in several applications, the implementation is use case dependent.
For this reason this work could be considered for poster presentation.